# Single-cell transcriptomics of human iPSC differentiation dynamics reveal a core molecular network of Parkinson's disease

Gabriela Novak [1,2,3✉], Dimitrios Kyriakis [1], Kamil Grzyb[1], Michela Bernini[1], Sophie Rodius[4], Gunnar Dittmar [4,5], Steven Finkbeiner [3] & Alexander Skupin [1,6✉]

Parkinson's disease (PD) is the second-most prevalent neurodegenerative disorder, characterized by the loss of dopaminergic neurons (mDA) in the midbrain. The underlying mechanisms are only partly understood and there is no treatment to reverse PD progression. Here, we investigated the disease mechanism using mDA neurons differentiated from human induced pluripotent stem cells (hiPSCs) carrying the ILE368ASN mutation within the *PINK1* gene, which is strongly associated with PD. Single-cell RNA sequencing (RNAseq) and gene expression analysis of a *PINK1*-ILE368ASN and a control cell line identified genes differentially expressed during mDA neuron differentiation. Network analysis revealed that these genes form a core network, members of which interact with all known 19 protein-coding Parkinson's disease-associated genes. This core network encompasses key PD-associated pathways, including ubiquitination, mitochondrial function, protein processing, RNA metabolism, and vesicular transport. Proteomics analysis showed a consistent alteration in proteins of dopamine metabolism, indicating a defect of dopaminergic metabolism in *PINK1*-ILE368ASN neurons. Our findings suggest the existence of a network onto which pathways associated with PD pathology converge, and offers an inclusive interpretation of the phenotypic heterogeneity of PD.

[1] The Integrative Cell Signalling Group, Luxembourg Centre for Systems Biomedicine (LCSB), University of Luxembourg, Esch-sur-Alzette, Luxembourg. [2] Luxembourg Institute of Health (LIH), Esch-sur-Alzette, Luxembourg. [3] Center for Systems and Therapeutics, the Gladstone Institutes and Departments of Neurology and Physiology, University of California, San Francisco, San Francisco, CA 94158, USA. [4] Department of Infection and Immunity, Luxembourg Institute of Health, Strassen, Luxembourg. [5] Department of Life Sciences and Medicine, University of Luxembourg, Belvaux, Luxembourg. [6] University of California San Diego, La Jolla, CA 92093, USA. ✉email: gabriela.novak@alumni.utoronto.ca; alexander.skupin@uni.lu

Parkinson's disease (PD) is one of the most prevalent neurological disorders, second only to Alzheimer's disease, with a prevalence of 1.8%, among persons over the age of 65 and 2.6% in the 85 to 89 age group[1–3]. As the average age of the population increases, PD is expected to pose an increasing burden to society. PD is characterized by the death of the midbrain dopaminergic (mDA) neurons found in the substantia nigra region of the brain, which are selectively sensitive to Parkinson's disease-associated neuronal cell death[4–7]. This results in the development of motor deficits, including bradykinesia, rigidity, and tremor, but many patients also develop non-motor symptoms, such as depression or dementia[8]. Unfortunately, current treatments only temporarily ameliorate the motor symptoms and do not reverse or slow down the progression of PD[4].

Most of our understanding of PD pathology at the molecular level is based on mutations known to cause PD, although these account for only 3–5% of PD cases, the remaining cases being idiopathic[2]. Despite the small fraction of cases they explain, these mutations provide an important window into the underlying molecular mechanisms of PD because they identify pathways which, when disrupted, are able to cause the disease. Many of these mutations converge on mitochondrial homeostasis, repair, and mitophagy. Hence, mitochondrial dysfunction likely plays a key role in the pathophysiology of PD[9]. An important group of these mutations lies within the *PINK1* gene. The PINK1 protein is expressed ubiquitously throughout the brain, in all cell types, where it localizes to the mitochondrial membrane[10]. PINK1 is a mitochondrial ubiquitin kinase and, together with the cytosolic ubiquitin ligase PARKIN, it targets damaged mitochondria for degradation via mitophagy, performing a mitochondrial quality control function needed to prevent accumulation of damaged mitochondria, which otherwise results in neuronal cell death[11–13]. The ILE368ASN mutation interferes with this process by reducing the interaction of PINK1 with its chaperone, HSP90, and destabilizing PINK1 at the mitochondrial membrane[11]. It also reduces its ubiquitin kinase activity through the deformation of its substrate-binding pocket and substrate misalignment[11]. Even though multiple publications have shown the involvement of PINK1 in mitophagy, its function is much broader. The targets of this kinase are involved in many cellular functions, including neuronal maturation[14], neurite outgrowth[15], suppression of mitophagy[16], and cell cycle modulation[17]. The broader impact on these pathways of loss-of-function mutations in this important kinase has not yet been fully elucidated[18].

One of the main obstacles to the study of Parkinson's disease is the death of mDA neurons and the resulting shortage of available postmortem samples. By the time of diagnosis, 60% of these neurons have disappeared and about 90% die by the late stage of the disease[6]. One approach is to study PD-associated mutations in animal models[19], but human-like mutations in animals often do not lead to the development of comparable pathology due to species differences in expression of key genes[20,21]. Thankfully, the development of cellular reprogramming allows nowadays for the conversion of somatic cells into induced pluripotent stem cells (iPSCs), which can subsequently be differentiated into neurons. This enables us to generate iPSCs from the skin cells of PD patients[22] and differentiate them into mDA neurons carrying disease-associated mutations[23–25]. Differentiating mDA neurons from iPSCs provides an almost unlimited source of neurons that allow for deep phenotyping and the elucidation of the cellular mechanisms underlying PD pathology.

Here, we generated iPSCs from the fibroblasts of a patient homozygous for the PD-associated mutation ILE368ASN (p.I368N) in the *PINK1* gene[2]. We used an optimized differentiation protocol to specifically generate mDA neurons, as this cell type displays a unique susceptibility to cell death in

PD;[23,25,26] the effect of PD on other types of DA neurons is variable[6,27].

The mDA neurons are unique and distinct from other DA neurons. Their development diverges from that of other DA neurons even before they commit to neural fate[28]. During early neural development, neural tube stem cells generate neurons and glia, the two basic building blocks of the brain. While other DA neurons follow this direct path, which is determined by the expression of the PAX6 transcription factor, mDA neurons develop from radial glial cells of the floor plate and their development is driven by early exposure to high levels of the SHH transcription factor[29], which prevents expression of the PAX6 transcription factor[30] and sets these cells on an entirely different developmental path[25,28]. As a result, mDA neuronal precursors follow a very unique signaling cascade, leading to the expression of a transcriptome that greatly differs from that of other DA neurons[21,25–28,31–35]. Their distinct identity is reflected in their function and current research indicates that this leads to their unique susceptibility to death in PD, which in turn has been directly associated with the classic movement symptoms of the disease[6,26–28,36]. This is also supported by the observation that gene expression differences between murine and human mDA neurons, which translate into subtle functional differences, lead to incomplete PD phenotype in animal models[19,21].

To investigate the disease mechanisms linked to the *PINK1* mutation, we performed extensive single-cell RNA sequencing (SC-RNAseq) analysis using Drop-Seq[37] at four different timepoints during mDA neuron differentiation[23–25]. We generated four pairs of samples, each consisting of a *PINK1*-ILE368ASN and a control (17608/6[38]) cell line differentiated in parallel. The pairs were differentiated in succession so that they would be at a different stage of differentiation on the collection day (Fig. 1). This also means that they represent four independent biological replicates. Pairwise differential expression analysis was then performed between the *PINK1*-ILE368ASN and control cell line of each pair, with a constraint that genes must be strongly and consistently dysregulated in all pairs, hence at all timepoints, to be considered in our analysis. The reasons for this are listed in the discussion section. Using databases of known protein-protein interactions, we show that these genes form a network and that its members directly interact with all 19 protein-coding PARK genes associated with PD. This suggests that other PD-associated mutations may also be acting through this common network of genes. Furthermore, the pathway most strongly associated with the genes of this network is the Parkinson's disease KEGG pathway. Subsequent proteomics analysis of differentiated cells confirmed the manifestation of the transcriptional modifications at the protein level. Our results point to the existence of a common disease mechanism that potentially underlies idiopathic PD and may represent a unifying perspective on PD progression that will guide future intervention strategies.

## Results

We performed a systematic differential expression analysis at a single-cell resolution between an iPSC line carrying the PD-associated ILE368ASN mutation in the *PINK1* gene and age- and sex-matched control cell line (control 1–2 in ref. [38]) during their parallel differentiation into mDA neurons (Fig. 1 and Supplementary Tables 1, 2). After preprocessing and quality-filtering, we used 4495 cells and 18,097 genes in our downstream analysis (Methods). For data integration, we performed a network analysis to identify the underlying key mechanisms of PD progression.

Fibroblasts were isolated from a 64-year-old male with PD symptom onset at 33 years of age who was homozygous for the ILE368ASN (P.I368N/P.I368N) mutation in the *PINK1* gene

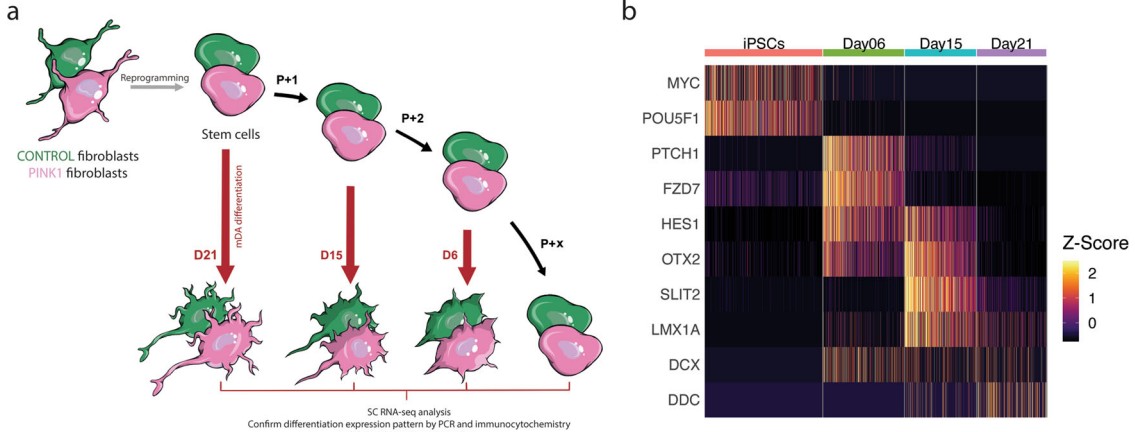

**Fig. 1 Experimental design. a** Fibroblasts were used to generate human induced pluripotent stem cells (iPSCs), which were then used to generate mDA neurons. Differentiation was initiated concurrently in a *PINK1*-ILE368ASN and a control cell line, at three different timepoints, to obtain cells which reach different stages of differentiation on the same collection day (generating four independent pairs). The samples were collected and processed for SC-RNAseq at the same time to avoid batch effects. "P + 1" indicates that the iPSCs were passaged before new differentiation was initiated. Since D10 was not used in the pairwise analysis, we indicated "P + 2" between D15 and D6 differentiation initiation. **b** Heatmap illustrating the transitions in gene expression from iPSC markers (*MYC* and *POU5F1-OCT3/4*), to genes associated with mDA differentiation (*PTCH1, FZDZ, HES1, OTX2, SLIT1*, and *LMX1A*), and finally to an early expression of mature mDA markers (*DCX* and *DDC*). This is discussed in more detail in the text and in Figs. 3 and 4. The gene expression matrix used here consists of 4495 cells (39,194 genes). Colors correlate to normalized counts (*z*-score, centered, and scaled) of the indicated gene.

(Coriell Institute, Cat. No. ND40066). The fibroblasts were confirmed to have a normal karyotype (Supplementary Fig. 1). Reprogramming was done at Yale Human Embryonic Stem Cell Core (New Haven CT) using the Sendai virus. The normal karyotype of iPSCs was confirmed (Supplementary Fig. 2 and Supplementary Tables 9, 10). Their iPSC status was ascertained by staining for the POU5F1 (also known as OCT4)[39–42] and the TRA-1-80[42,43] iPSC markers (Fig. 2a), by expression of key iPSC status markers using sc-RNAseq (Fig. 2b), and by the TaqMan iPSC Scorecard Assay[44,45], which also confirmed the trilineage potential of the cell line[44] (Fig. 2c).

**Single-cell RNAseq (sc-RNAseq) analysis reveals gene expression panel for direct classification of iPSCs' stemness or pluripotency.** Staining for OCT/TRA proteins and Scorecard are common approaches for determining iPSC status (reviewed by Smith and Stein)[42]. Here we show that a panel of genes indicative of iPSC status is readily detectable by single-cell analysis and can be used to indicate iPSC status directly in the cells used in an sc-RNAseq experiment, rather than by staining or expression analysis of an independent sample, which in some cases may not reflect the iPSC status of the experimental sample. Furthermore, this may be useful in cases where the samples are no longer available, such as for data obtained from an sc-RNAseq data repository. In our dataset, we quantified the expression of genes commonly used to ascertain iPSC status (*MYC*[46] and *POU5F1*[39–42]) and showed that these can be readily detected by sc-RNAseq analysis (Fig. 2b). However, sc-RNAseq analysis comes with some limitations. In particular, it is not able to detect genes which are naturally expressed at low levels. We, therefore, created a list of genes associated with high stemness, i.e., expressed selectively in iPSCs exhibiting full stem cell phenotype, which are readily detectable in sc-RNAseq data, creating a link between an iPSC state characterized by standard techniques and a signature visible in sc-RNAseq data. The heatmap of top genes differentially expressed during the transition between iPSC and subsequent differentiation stages shows that the iPSCs express several genes associated with stemness (Fig. 2b). For instance, we detected the expression of *TDGF-1*, which was shown to be expressed by stem cells with the highest expression of stemness

markers[41]. Additional genes expressed by the iPSCs were *L1TD1*, *USP44*, *POLR3G*, and *TERF1* (essential for the maintenance of pluripotency in human stem cells[47–50]), as well as *IFITM1*, *DPPA4*, and *PRDX1* (associated with stemness[51–53]).

Based on our observations, we propose that the following panel of genes should provide a reliable indication of stemness in single-cell experiments: *MYC (cMyc), POU5F1 (Oct4), LIN28A, TDGF-1, L1TD1, USP44, POLR3G*, and *TERF1* (Fig. 2b).

**In vitro differentiation of iPSC-derived mDA neurons recapitulates the in vivo process.** As stated by Bjorklund & Dunnett "expression of TH is not in itself sufficient to prove that a neuron is catecholaminergic, let alone dopaminergic"[35]. Hence, we made great effort to confirm that the neurons generated by our protocol display a true mDA phenotype.

To confirm that our differentiation protocol (Supplementary Table 1) recapitulates the in vivo mDA differentiation path, we identified genes that are essential and specific to the in vivo mDA differentiation process (*OTX2, EN1, LMX1B, LMX1A, FOXA2, MSX1, NR4A2, PITX3*, and others) (Supplementary Table 3)[25–28,33–35] and analyzed their expression during the development of the control cell line at timepoints D0 (iPSCs), D6, D10, D15, D21 D26, D35, and D50, which represent the major developmental steps of the protocol (Fig. 3).

We first imaged cells at timepoints Day 25 and Day 35, as at this stage the cells should have developed mDA characteristics. Staining for key mDA protein markers TH, PITX3, LMX1A, and DAT, with MAP2 as a neuronal marker, confirmed the mDA phenotype (Fig. 3a). (The co-expression of these mDA markers with TH is shown in Fig. 4a.) At D25, neuronal cells possess only short processes and generally low mDA marker expression, but by D35 their axons are far longer and mDA marker expression is more defined and more robust. The mRNA expression of *TH, LMX1A*, and *ALDHA1A* was further validated by qPCR, and the trajectory of these genes' expression indicated the development of mDA characteristics by Day 21 (Fig. 3b), in agreement with the imaging results at Day 25 (Fig. 3a).

To characterize the differentiation process in more detail, we performed the sc-RNAseq analysis at eight timepoints of the differentiation process. Analysis of differentially expressed genes

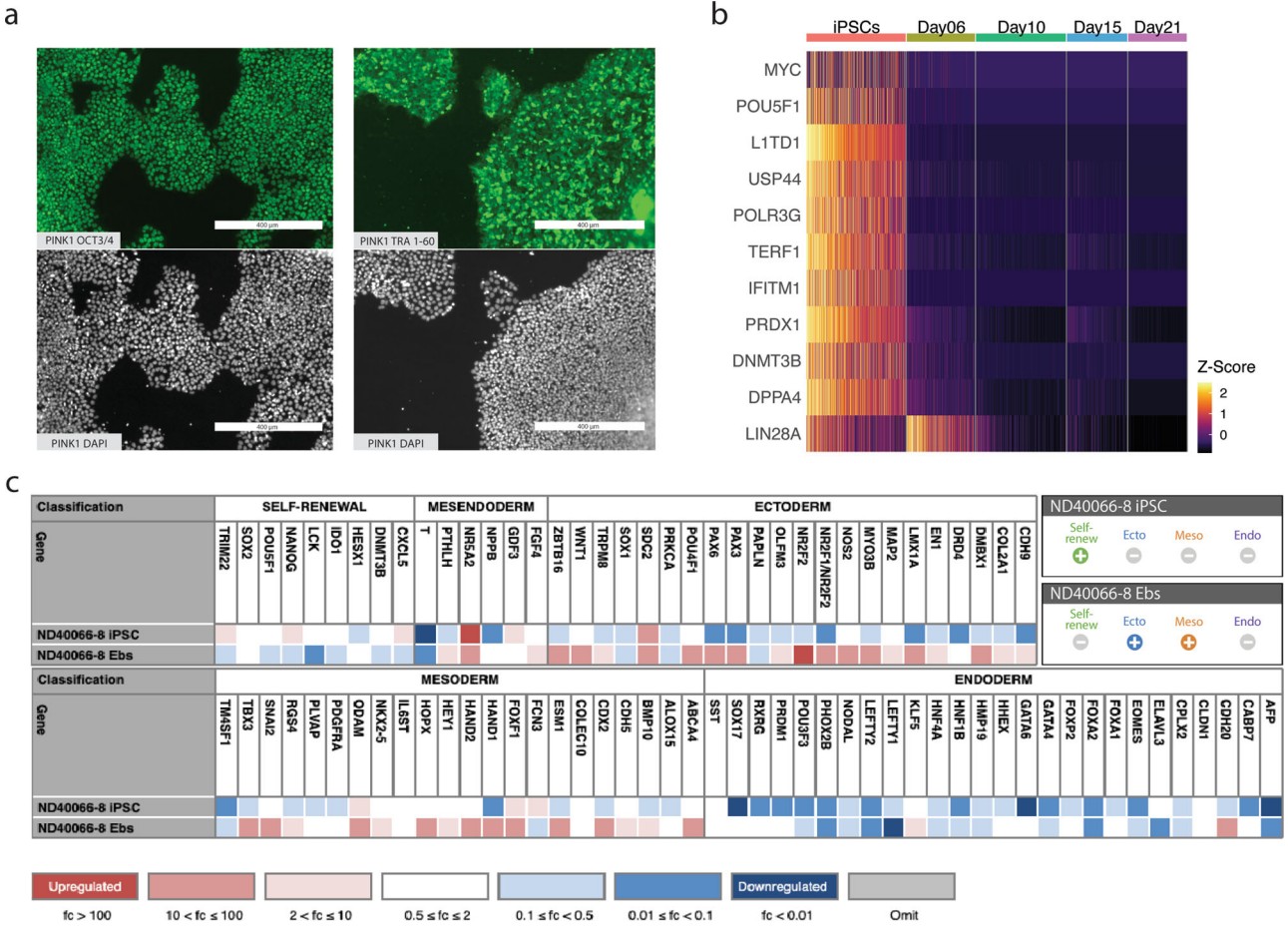

**Fig. 2 Classification of iPSC status. a** Immunocytochemistry (ICC). Staining for the iPSC markers POU5F1 (more commonly known as OCT3/4) and TRA-1-60 of iPSC colonies, prior to differentiation. DAPI was used to stain cell nuclei as a reference. **b** Expression of genes known to indicate iPSC status (*MYC* and *POU5F1*) and of genes identified by a differential expression analysis between iPSCs and differentiating cells (also see Supplementary Fig. 12). Colors correlate to normalized counts (*z*-score, centered, and scaled) of the indicated genes. *TDGF-1* is expressed in iPS cells of high stemness;[41] *L1TD1, USP44, POLR3G,* and *TERF1* are essential for the maintenance of pluripotency in human stem cells;[47-50] *IFITM1, PRDX1, DNMT3B, DPPA4,* and *LIN28A* and are associated with stemness[51-53,137,138]. **c** Results of Scorecard analysis of iPSCs and embryonic bodies (EBs)[44,45]. iPSCs are expected to show high expression of self-renewal genes and low expression of mesoderm, ectoderm, and endoderm markers. EBs are cells at an early stage of spontaneous differentiation. Scorecard analysis of EBs determines the iPSC line's potential to differentiate into the three germ layers, hence, EBs are expected to express few or no self-renewal genes and to show expression of some mesoderm, ectoderm and endoderm markers: Ecto±, Meso±, Endo±.

across timepoints revealed the expression of specific differentiation stage modules (Supplementary Fig. 12), in accordance with known in vivo stage-specific expression patterns (Fig. 3c and Supplementary Table 3). For example, in vivo, the development of mDA phenotype depends on the early high expression of Sonic Hedgehog (SHH), followed by the induction of Wnt signaling and the expression mDA-specific downstream pathways[25,26,28] (Fig. 3c and Supplementary Table 3). Consistent with these in vivo differentiation steps, *PTCH1*, a receptor for *SHH*, and *FZD7*, a receptor for Wnt proteins (Fig. 3c) were among the highest-expressed genes on day 6 (D6) of the differentiation protocol. The presence of *EN1* as a key entity was confirmed by qPCR (Supplementary Fig. 15), as its expression level was too low for detection by sc-RNAseq. The sc-RNAseq analysis again revealed that at Day 21 many factors that are specific to the mDA differentiation path, such as *TCF12, ALCAM, PITX2, ASCL1,* and *DDC*[27,54-57], were among the most highly expressed genes (Fig. 3c and Supplementary Fig. 12). Overall, these observations confirm that our in vitro differentiation protocol does indeed recapitulate the in vivo differentiation of mDA neurons and produces genuine mDA neurons (PAX6-, ALDH1A1+, PITX3+, KCNH6/GIRK2+, NR4A2/NURR1-, and LMX1A+), rather

than other types of DA neurons (PAX6+ and ALDH1A3+) (Table 1 and Supplementary Fig. 3).

However, this assessment of the differentiation process of human mDAs was mostly based on the pattern of mDA differentiation gene expression in murine brains. We, therefore, compared our data with the recently outlined pattern of gene expression during human mDA[21,58]. The pattern of gene expression during our in vitro differentiation of human iPSCs into mDA neurons matched the pattern of human mDA differentiation[21] more closely than that of murine neurons, confirming the validity of our differentiation protocol.

Using the gene expression groups associated with different stages of maturation, from radial glia (Rgl), progenitors (Prog), to neural progenitors (NProg), and finally to mDA neurons (DA)[21] (Supplementary Table 3), we could characterize the differentiation trajectory by the level of gene expression (Fig. 3d). We then used these gene groups to characterize individual cells with respect to their most likely cell type and determined the population dynamics by the percentage of cell types present at each timepoint (Fig. 3e). The analysis showed fast differentiation from iPSC state to a neuronal lineage by Day 6, and the subsequent maturation towards an mDA phenotype starting at

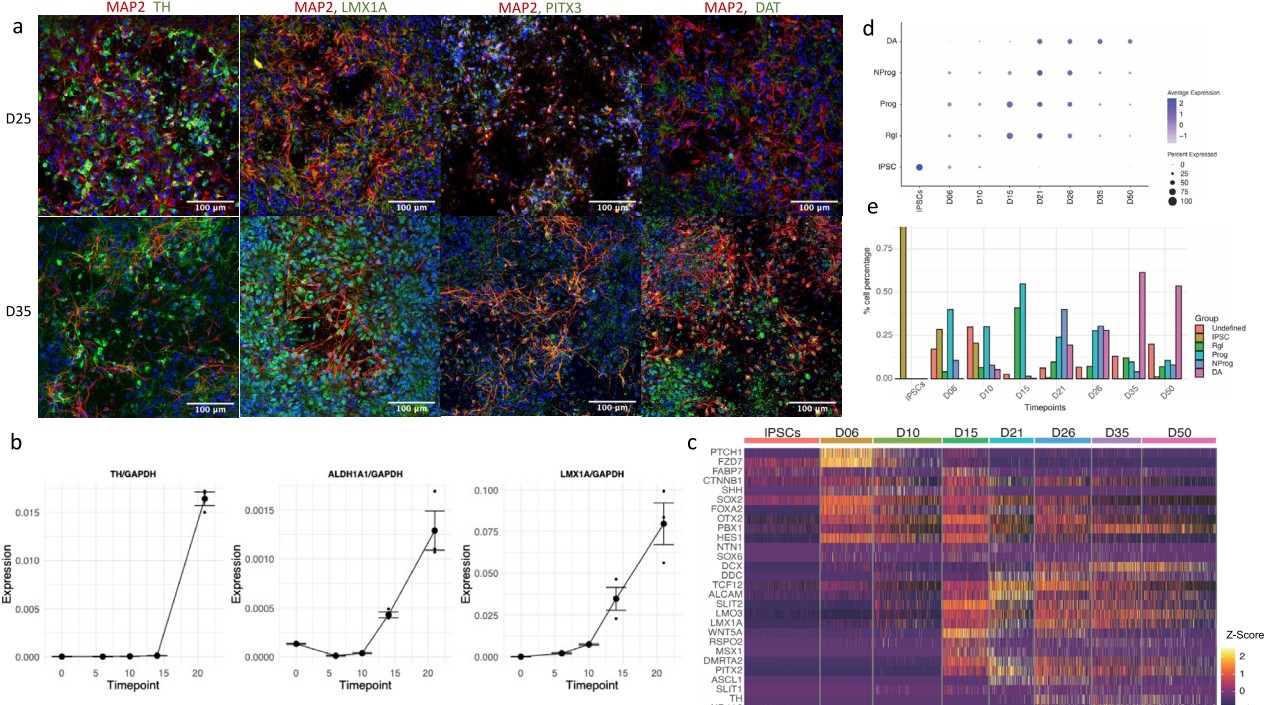

**Fig. 3 In vitro differentiation of iPSC-derived mDA neurons recapitulates the in vivo process. a** To illustrate the maturation of neuronal morphology and mDA status, differentiated neurons were stained at D25 and D35 for a neuronal marker MAP2 (red) and mDA markers (green): TH, PITX3, LMX1A, and DAT. While D25 neurons show short processes and low expression of mDA markers, D35 neurons show much longer axons and well-defined expression of mDA markers (green/red overlap resulting on orange/yellow). **b** Quantitation of mDA markers *TH, ALDH1A1,* and *LMX1A,* using absolute quantitation via qPCR. Each timepoint represents three independently differentiated biological replicates, amplified in duplicate. Standard error (SE) bars are the SE of biological replicates. The expression levels are standardized to total RNA and to the expression of the housekeeping gene *GAPDH* (see Methods). **c** Heatmap showing the expression of genes known from the literature to be involved and necessary for mDA neuron differentiation (Supplementary Table 3). Colors correlate to normalized counts (z-score, centered, and scaled) of the indicated genes. **d** The mDA differentiation gene expression profile recently published by Ásgrímsdóttir and Arenas (2020)[21] was used to show the progression during differentiation, from iPSCs to radial glia (Rgl), to progenitors (Prog) and neuroprogenitors (NProg), and to early mDA neurons (DA). Genes used to determine the expression modules are listed in Supplementary Table 3. **e** Proportions of cells expressing the various phenotypes illustrated in (**d**). The gene expression matrix obtained by SC-RNAseq used here consists of 4495 cells (see Methods section).

Day 21, accompanied by the increasing prevalence of DA phenotype, from 20% at Day 21, to 28% at Day 26, and 61% at D35, after which it seemed to stabilize (Fig. 3d). This characterization further confirms that early mDA differentiation is achieved around Day 21.

**The PINK1-ILE368ASN mutation is associated with persistently dysregulated expression of nearly 300 loci.** To investigate the effect of the *PINK1* mutation on mDA development, we differentiated the control and the *PINK1*-ILE368ASN cell lines in parallel (Figs. 1, 4) and focused on the early differentiation period, to increase our chances of finding the direct effects of *PINK1*-ILE368ASN, as described below. Co-staining of TH positive neurons with the midbrain dopaminergic markers PITX3, LMX1A, and DAT in both the control and PINK1 cell lines identified neurons at day D21 as early postmitotic mDA neurons[25] with clearly neuronal morphology and no major differences between the cell lines (Fig. 4a).

To investigate potential underlying mechanisms of the PINK1 mutation, differential expression between the two, in parallel differentiated, cell lines at each timepoint was determined and genes that were identified as differentially expressed at all four timepoints were identified. Each timepoint is an independent biological replicate, initiated at a different time and with cells of a different passage number. Control and *PINK1*- ILE368ASN cells

co-clustered together based on their differentiation stage, from iPSCs, to day 6 (D6), D15, and D21 (Fig. 4b), indicating that overall RNA expression was specific to differentiation stages, and rather uniform between cell lines, which was amenable to the identification of subtle gene expression differences due to the presence of a mutation in the *PINK1*- ILE368ASN cell line.

The *PINK1*- ILE368ASN cells at D10 showed low viability, hence the D10 timepoint was not included in the pairwise analysis. After preprocessing and quality-filtering (Methods and Supplementary Fig. 4), a total of 4495 cells (2518 control and 1977 PINK1 cells) and 18,097 genes were included in our analysis. UMAP analysis of the single-cell data revealed that gene expression was rather similar between the cell lines and mainly defined by differentiation stage, rather than by cell line origin (Fig. 4b). In accordance with the staining results (Figs. 3a, 4a), we observed the onset of expression of the mature mDA markers *TH* and *KCNH6* (also known as *GIRK2*) on D21 (Fig. 4c).

The analysis of pairwise differential expression at each timepoint (adjusted $p$ values ($p_{adj}$) <0.01 fold changes (FC) >0.1) (Fig. 5a) identified 14 genes that were upregulated and 13 genes that were significantly downregulated in the *PINK1*-ILE368ASN cell line compared to control (Fig. 5b and Table 2, indicated by "X"). Because iPSCs are very different from differentiating neuronal precursors, we next tested whether including iPSCs had disproportionately affected the results by excluding neuron-specific genes. Repeating the analysis on D6,

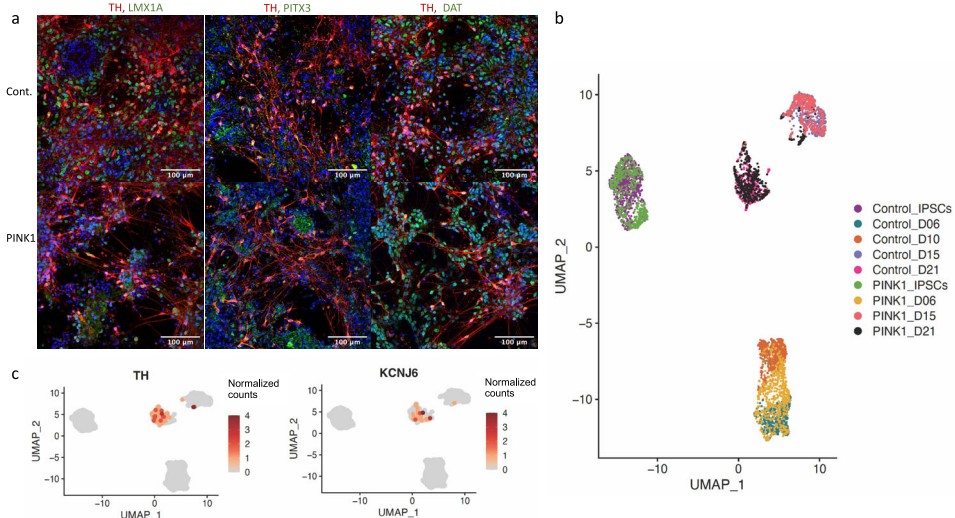

**Fig. 4 Classification of mDA status. a** TH positive neurons co-express mDA markers PITX3, LMX1A, and DAT in control (top) and PINK1 cell line (bottom), at D35. For images of individual targets see Supplementary Fig. 13 and for colorblind-friendly images see Supplementary Fig. 14. **b** Based on our SC-RNAseq data, cell lines cluster according to differentiation stage, indicating that gene expression is very homogenous between the control and the *PINK1*-ILE368ASN cell lines, which allows for the detection of even subtle alteration induced the presence of the PINK1 mutation. **c** Trajectory of expression of *TH* and *KCNJ6* (*GIRK2*), two mDA neuron markers. At D21 neurons begin to show *TH* expression, together with an expression of other mDA markers, which indicates that they are becoming early postmitotic mDA neurons[25]. Similar observations can also be made from an expression heatmap shown in Supplementary Fig. 12. The scale represents normalized counts.

**Table 1 DA neuron heterogeneity: mDA and non-mDA markers.**

| Dopaminergic neuron type | TH | DDC AADC | SLC6A3 DAT | SLC18a2 VMAT | PAX6 | Other | |
|---|---|---|---|---|---|---|---|
| A8–10 midbrain dopaminergic neurons | + | + | + | + | - | **ALDH1A1** | |
| A11 periventricular nucleus (PVN) | + | + | - | + | - | **ALDH1A3** | |
| A12 arcuate nucleus (endocrine) | + | + | + | + | - | **ALDH1A3** | Dlx1 |
| A13 medial zona incerta | + | + | - | + | +★ | **ALDH1A3** | Dlx1, SST |
| A14 preoptic periventricular nucleus | + | + | - | + | +★ | **ALDH1A3** | |
| A15 preoptic & endopeduncular area | + | - | - | +? | + | **ALDH1A3** | |
| A16 periglomerular cells, olfactory bulb | + | + | + | - | + | **ALDH1A1** | |
| A17 interplexiform cells in the retina | + | + | + | | | **NKR** | |

When studying PD, it is important to ascertain that the DA neurons are in fact mDA neurons. In an in vitro differentiation system, simple marker combinations that normally distinguish mDA neurons, such as positional and anatomical information, are missing. We relied instead on molecular markers culled from the literature to monitor our differentiation protocols. (★ A13 and A14 PAVH express PAX6 transiently during development. PAX6 is expressed early in development, whereas the remaining markers are expressed later and are markers of mature DA neurons. "?" indicates that the literature regarding the expression is not unanimous.)[25,34,35,139-144].

D15 and D21 only identified 28 genes that were upregulated and 27 genes that were downregulated at all three timepoints, including all genes previously identified (Table 2). As expected, excluding iPSCs resulted in the identification of a broader range of genes because genes that are differentially expressed only in the neuronal lineage were previously excluded due to the requirement that DEGs be dysregulated at all timepoints. However, both sets are equally valuable, as genes dysregulated even in iPSCs are likely to participate in systemic PD pathology, regardless of cell type, and may be relevant to a broader spectrum of PD pathology than the death of mDA neurons. Interestingly, most of the differentially expressed genes are already linked to PD, other PD mutations, or neurodegeneration (Table 2).

For an alternative definition of differentially expressed genes (DEGs), we used the maximum adjusted *p* value in a pairwise combination as an adjusted *p* value, and the average fold change that occurred in the pairwise comparison as a fold change threshold. With this approach, we retained only genes dysregulated in the same direction at all timepoints. This analysis led to 151 DEGs (named Group B, Supplementary Table 4), which included the previously identified genes of Group A, and of which 65 were upregulated and 86 downregulated compared with controls ($p_{adj} < 0.01$ and FC > 0.1). Taking the mean of FC of the different timepoints enhanced the identification of DEGs because it reduced the effect of the variability between pairs due to their different differentiation states. Repeating the same analysis for the four timepoints (iPSCs, D6, D15, and D21), but taking into account only the absolute degree of change in iPSCs, yielded 172 genes (Group C, Supplementary Table 5). Repeating the analysis using only timepoints D6, D15, and D21 identified a total of 286 DEGs (Group D) (also see Fig. 6a and Supplementary Data 1). Together, when all analyses were pooled, we obtained 292 DEGs (six genes in Group C depended on the inclusion of iPSCs and did not appear in Group D, see Supplementary Data 1).

**Enrichment analysis reveals a strong association with the KEGG Parkinson pathway.** Enrichment analysis was performed using the STRING[59] database (Fig. 5c). The highest-associated KEGG pathways were the Parkinson's, Huntington's, and spliceosome pathways. Details are listed in Supplementary Data 5. Biological processes most strongly associated with the DEGs were C3HC4-type RING finger domain binding, Ran GTPase binding, and protein

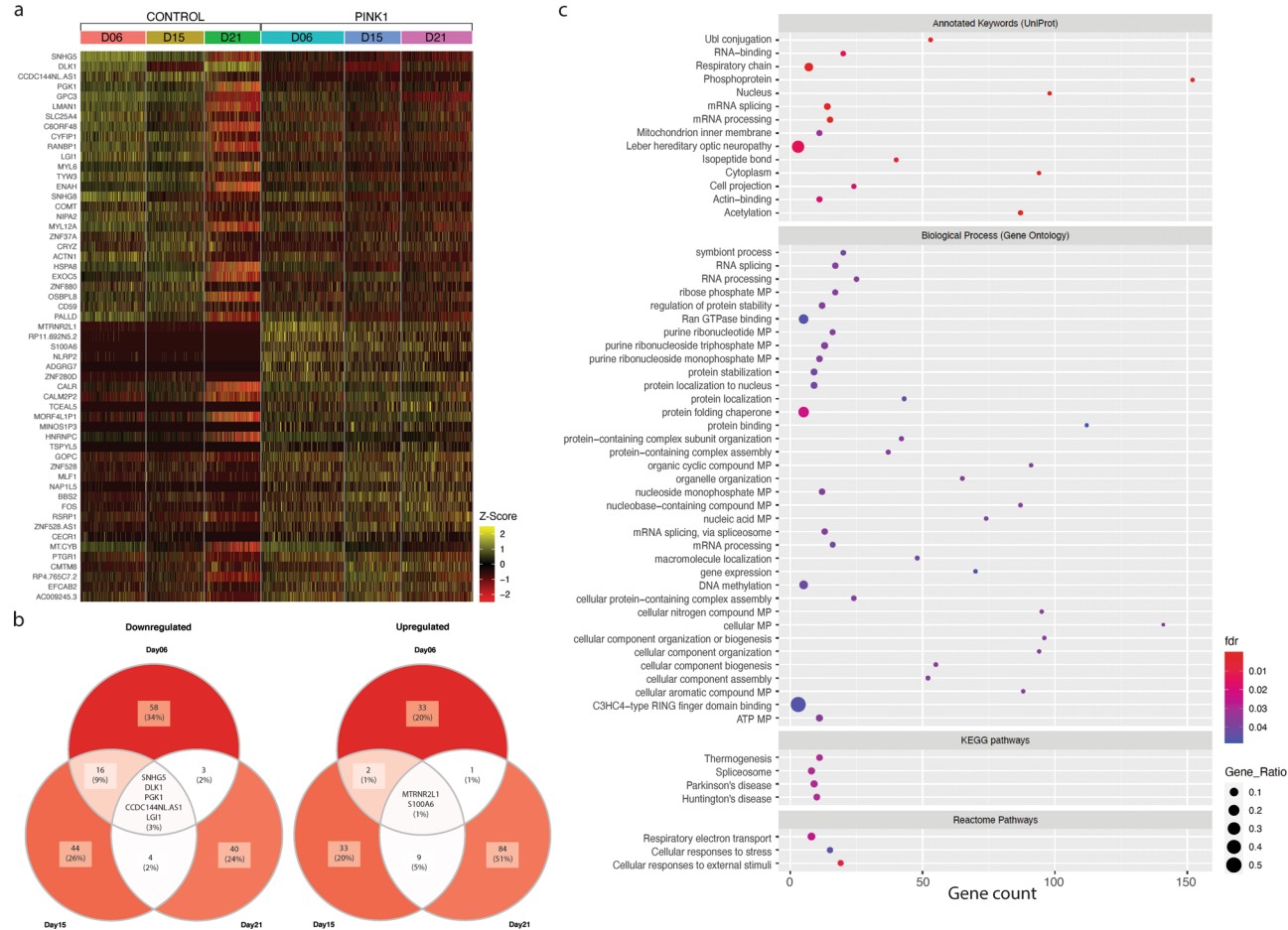

**Fig. 5 Differentially expressed genes (DEGs) in a cell line homozygous for a mutation in the *PINK1* gene, compared to a control cell line, at three timepoints during the differentiation of mDA neurons (D6, D15, and D21). a** Heatmap of the top DEGs. Each column corresponds to a timepoint for either control or PINK1 cells; each row shows the expression of one gene in individual cells. Colors correlate to normalized counts (z-score, centered, and scaled) of the indicated genes. **b** Top DEGs. The minimum fold change was increased to highlight the top differentially expressed genes. We identified the top 56 genes as our group A (Table 2); here we show the top five upregulated genes (left Venn diagram) and the top three downregulated genes (right Venn diagram). **c** Enrichment analysis performed using the STRING[59] database. The top KEGG pathway associated with this dataset is Parkinson's disease. The other three KEGG pathways identified were Spliceosome, Huntington's disease, and Thermogenesis. Details are listed in Supplementary Data 5. The gene expression matrix used for the downstream analysis consisted of 4495 cells (39,194 genes) and differential expression analysis resulted in 292 DEGs, which were used to perform the enrichment analysis.

folding chaperones. Respiratory chain transport was the most strongly associated Reactome pathway.

**Data integration reveals a common PD network**. To integrate the expression analysis and identify underlying disease mechanisms, we generated a network of interactions between the DEGs via Gephi[60], using protein–protein interaction (PPI) information obtained from the STRING and GeneMANIA databases[59,61]. The network we obtained includes 246 of the 292 DEGs, since pseudogenes and non-coding RNAs could not be integrated into a protein–protein interaction network (Fig. 6), and 2122 interactions (Supplementary Data 2). The curated network contains only DEGs and any genes that were automatically added by the databases were removed to ensure a reliable core network based solely on DEG data. Based on known protein–protein interactions, the DEGs integrate into a close-knit core network in which several DEGs form central nodes (Fig. 6). To evaluate the importance of the DEG-based PPI network produced by STRINGdb (v10)[59], we compared the DEG-based network with corresponding random networks generated from sets of 292 randomly chosen genes excluding DEGs. Based on 50 random

networks, we show that the DEG-based network includes significantly more protein-coding genes and interactions than by chance (Supplementary Fig. 5) and that the network structure in terms of degree distribution is significantly distinct as evaluated by the Wilcoxon test ($p = 2.22$e-16) and indicates the mechanistic character of the network.

The network of genes dysregulated by the presence of the *PINK1-ILE368ASN* mutation includes genes related to other PD-associated pathways, which is intriguing since it was generally assumed that each PD-associated mutation leads to PD pathology via an independent, characteristic path. For example, two DEGs, *GOPC*, and *GPC3*[62,63] interact with the PD-associated gene *DJ-1* (*PARK7*)[2,64]. The DEG network also includes genes of the *LRRK2* (*PARK8*) network[2,64], namely *ENAH, HSPA8, MYL6, MALAT1*, and *SNHG5* (Supplementary Fig. 6). *SNHG5* and *MALAT1* interact with *LRRK2* via *miR-205-5p*[44,45]. DLK1 and MALAT1 mediate α-synuclein accumulation[65,66]. In fact, the DLK1-NURR1 interaction involved in this process may be mDA neuron-specific[67], highlighting the necessity to use mDA neurons for the study of PD-related pathways. Additionally, MALAT1 was shown to increase α-synuclein protein expression[68]. In short, this suggests that interactions leading to PD pathology are more

**Table 2 The top genes dysregulated consistently in PINK1 vs. control cells across differentiation stages.**

| Upregulated in PINK1 | | | | | Downregulated in PINK1 | | | | |
|---|---|---|---|---|---|---|---|---|---|
| GENE | incl. iPSCs | excl. in STRING | PD association | Ref. | GENE | incl. iPSCs | excl. in STRING | PD association | Ref. |
| AC009245.3 | X | Pseudogene | | | ACTN1 | | | PD | 145 |
| ADGRG7 | X | | rare var., mito | 146 | C6ORF48 | X | | | |
| BBS2 | | | | | CCDC144NL.AS1 | X | RNA | | |
| CALM2P2 | X | Pseudogene | | | CD59 | X | | PD | 147 |
| CALR | | | PD | 148 | COMT | | | PD | 149 |
| CECR1 | | | | | CRYZ | X | | GWAS, PD Gene | 63 |
| CMTM8 | | | GWAS, PD | 63 | CYFIP1 | X | | (via mTOR) | 150 |
| EFCAB2 | X | | microarray | 151 | DLK1 | | | PD | 152 |
| FOS | | | rat, L-DOPA | 153 | ENAH | | | GWAS, LRRK2 | 154 |
| GOPC | | | PARK7 (DJ-1) | 63 | EXOC5 | | | Parkinson Dis.Map | 155 |
| HNRNPC | X | | binds Parkin | 101 | GPC3 | | | reduced in DJ-1 mut. | 62 |
| MALAT1 | | | PD | 156 | HSPA8 | X | | PD, LRRK2 | 157 |
| MINOS1P3 | X | Pseudogene | | | LGI1 | | | PD | 119 |
| MLF1 | | | via HTRA2 | 158 | LMAN1 | | | Parkin transloc. | 104 |
| MORF4L1P1 | X | Pseudogene | | | MYL12A | | | binds Parkin | 101 |
| MT-CYB | | | mito | 159 | MYL6 | X | | interacts with LRRK2 | 160 |
| MTRNR2L1 | X | | binds Parkin | 101 | NIPA2 | X | | tremor | 161 |
| NAP1L5 | | | | | OSBPL8 | | | via ZNF746, Biogrid | 162 |
| NLRP2 | X | | inflammasome | 163 | PALLD | | | PD | 164 |
| PTGR1 | | | | | PGK1 | X | | PD | 165 |
| RP11.692N5.2 | X | Pseudogene | | | RANBP1 | | | | |
| RP4.765C7.2 | X | Pseudogene | | | SLC25A4 | | | binds Parkin | 101 |
| RSRP1 | | | | | SNHG5 | X | RNA | via miR-205, LRRK2 | 166 |
| S100A6 | X | | PD | 167 | SNHG8 | | RNA | | |
| TCEAL5 | X | | | | TYW3 | X | | | |
| TSPYL5 | | | Ubiquit. | 168 | ZNF37A | X | | | |
| ZNF280D | X | | GWAS* | 169 | ZNF880 | X | | | |
| ZNF528 | | | | | | | | | |
| ZNF528.AS1 | | RNA Gene | | | | | | | |

Pairwise differential expression analysis of each timepoint (iPSCs, D6, D15, and D21), resulted in 14 genes that were upregulated and 13 genes that were downregulated in the PINK1-ILE368ASN cell line, compared to control (p_val_adj <0.01 and abs(avg_logFC) >0.1); these genes are marked with "x" in column "Incl. iPSCs". Twenty-nine additional genes were identified in an analysis that included D6, D15, and D21, but not iPSCs. The "Excluded" column explains why a gene was not included in the protein–protein interaction network. These 56 top DEGs are later referred to as Group A. The gene expression matrix used for the downstream analysis consisted of 4495 cells (39,194 genes). * rs11060180.

complex than one mutation - one path to PD, as generally thought. It also indicates that many druggable targets may be useful in treating PD and that these may be universally effective for PD caused by several different mutations, and perhaps even for idiopathic PD. For example, terazosin, which is already in clinical use, was found to be associated with slower disease progression, likely by enhancing the activity of phosphoglycerate kinase 1 (PGK1)[69], one of the top DEGs identified in our study.

For the evaluation of the relative importance of each node within the network, we applied betweenness centrality[60] (Fig. 6a), an approach that reveals the overall connectedness of each gene. Genes onto which several other genes converge are shown as large circles or nodes, their size being proportional to the number of interactions they form. Interestingly, the major nodes of this network are genes already known to play an important role in ubiquitination (Fig. 6b) and PD pathology (Fig. 6c and Table 3). Next, we built a correlation network (p value < 0.05, r > 0.1) of the 246 DEGs based on the normalized counts. By extracting the common interactions of these two networks, we obtained a network with 297 interactions (Supplementary Table 6), which highlights protein–protein connections that correlate with differential expression of the genes. This analysis further supports the role of the connections between these genes in mediating the resulting differential expression in the presence of the PINK1-ILE368ASN mutation. STRING was subsequently used to high-light functional pathways represented within the DEG network (Supplementary Fig. 7 and Supplementary Data 3). Several

pathways known to play a role in PD pathology are strongly represented within the network, notably ubiquitination[19,70], mitochondrial pathways[9,71], cellular response to stress[72], lysosomal proteins[73], protein metabolism (localization, modification, transport, folding, and stability), RNA processing[74], aromatic compound metabolism[75–78], vesicle-mediated transport and exocytosis[79], and cellular catabolic processes[72,73] (Supplementary Fig. 7). Importantly, the strongest-associated pathway is the KEGG-PD[59] pathway (Supplementary Fig. 9a). The CHCHD2 gene was identified as a dysregulated gene through our analysis, but it was also recently identified as a PD-associated gene and named PARK22[64,80,81] (Fig. 6a).

To investigate further how the identified network relates to other known PD mechanisms, PD-associated genes, also known as PARK genes (Supplementary Table 6 and Supplementary Fig. 9), were added to the DEG network. Next, PARK–PARK interactions were removed and only PARK–DEG interactions were retained to test how PARK genes integrate into the network. All 19 protein-coding PARK genes[2,64] interact directly with at least one, but usually several DEGs (Supplementary Fig. 9). The degree of interaction of PARK genes with the DEGs of the network is illustrated by coloring (in pink) DEGs that directly interact with a PARK gene. The darker the color, the greater the number of PARK genes the DEG interacts with. The preexisting central nodes of the network generally interact with several PARK genes, suggesting that they play a central role in linking the PARK genes to the network, but also that PARK genes may mediate PD

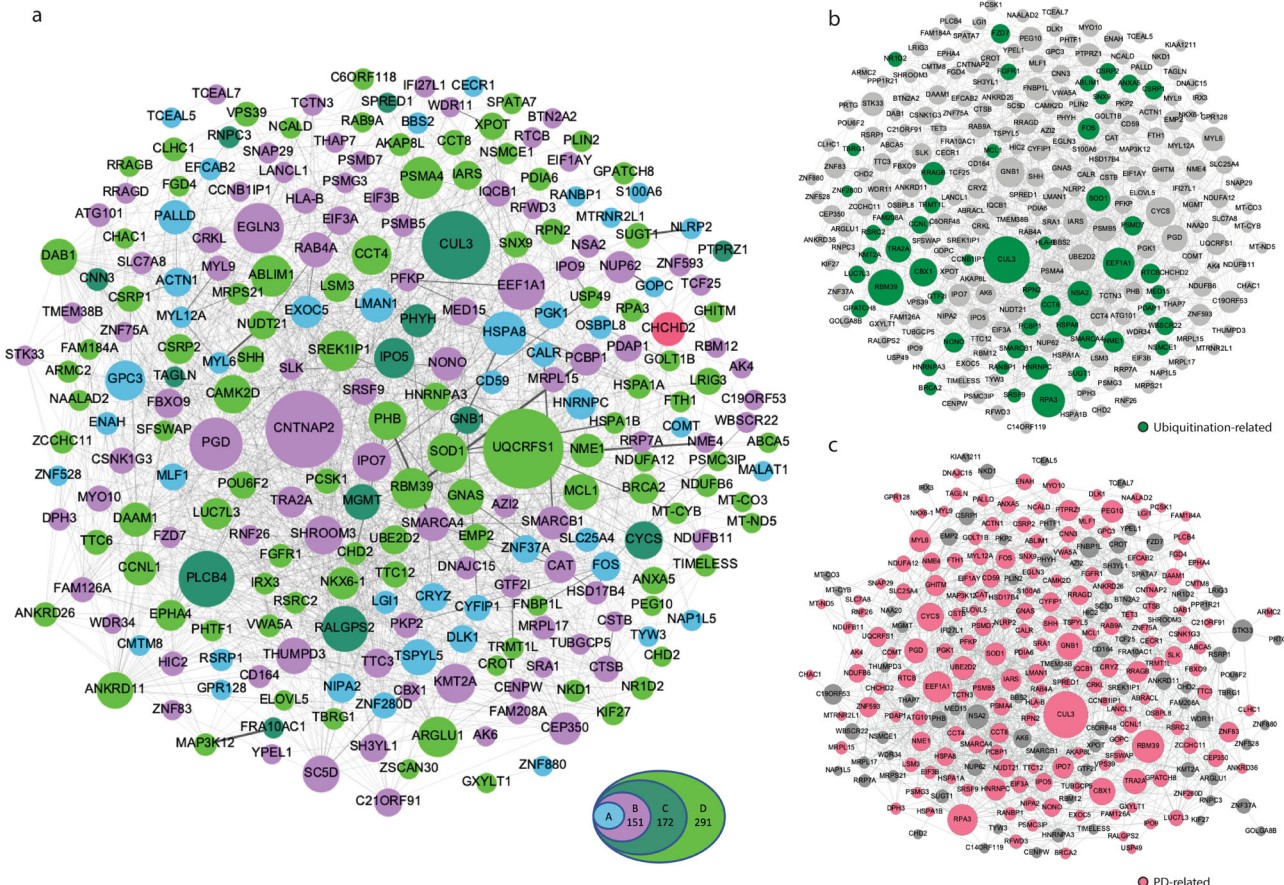

**Fig. 6 Network analysis. a** Protein–protein interaction network based on known interactions available through the STRING[59] and GeneMANIA[61] databases. Only strong interactions were retained, predicted interactions or text associations were omitted (see Methods). Betweenness centrality was used to illustrate the relative importance of each node within the network through the level of its connectedness to other proteins. The larger the circle, the more partners the node is connected to. The colors represent the four DEG sets, with the top 56 DEGs (group A) in light blue, group B in purple, group C in dark green, and group D in light green. Each set consists of genes of the previous set plus additional genes identified by the new parameters. *CHCHD2* (pink, part of group B) is a DEG, which has recently been identified as a PARK gene. Random selection of genes from genes detected by sc-RNAseq did not lead to a network formation (Supplementary Fig. 5). **b** DEGs which play a role in ubiquitination. Additional functional pathways are listed in Supplementary Fig. 7 and Supplementary Data 3. Specific connections to ubiquitination proteins are shown in Supplementary Fig. 8. **c** Based on the literature, 68% of the DEGs of this network are already known to be associated with PD (for references see Supplementary Data 4). Supplementary Fig. 9 shows which genes/proteins of the network directly interact with PARK genes through known protein–protein interactions. The topology of all three networks is the same, the different appearance is a result of a separate analysis run, but the connections and size of the nodes remain identical.

pathology through a few central pathways of this network, and that the effects of different PARK genes converge on the same set of pathways (Supplementary Fig. 9).

Further analysis revealed that a large number of the DEGs interact with genes associated with mitochondria or ubiquitination (Fig. 6b and Supplementary Fig. 8). For this analysis, we used BioGRID[61,82] to identify interactions with mitochondrial or ubiquitination proteins for the top 172 DEGs (groups A–C). These interactions were used to create a network illustrating that many of the DEGs in our study directly interact with genes involved in mitochondrial function and in ubiquitination. Only direct DEG to mitochondrial gene or DEG to ubiquitination gene interactions were included and PARK genes were added for reference (Supplementary Fig. 8). Based on manual literature search, we determined that at least 68% of the DEGs (174 of 255 genes, not including pseudogenes and RNA genes) are already directly associated with PD, either experimentally, or linked through GWAS-PD, or by PD expression studies (Fig. 6c and Supplementary Data 4). This is particularly true for the major nodes of the network (Table 3 and Fig. 6c).

**Proteomics analysis confirms impaired neuronal phenotype in *PINK1*-ILE368ASN mutant line.** To investigate how the identified transcriptional modifications manifest in the neuronal phenotypes, we performed proteomics analysis at an early (day 25) and later maturation stage (day 40). The analysis identified 39 differentially abundant proteins in *PINK1*-ILE368ASN cells as compared to controls, based on biological duplicates with a log2 fold change larger than 1 (Fig. 7a). Of these, four differ at both timepoints (D25 and D40). Overall, 31 proteins were differentially abundant at D25, including CSRP2 and VWASA, which were also identified by sc-RNAseq as differentially expressed at the mRNA level at D6, D15, and D21 (Fig. 7b and Supplementary Table 8). At D40, 12 proteins were found to be differentially abundant, including four also identified at D25, namely TH, DDC, NES, and VIM. We performed a network analysis based on the differentially abundant proteins (Fig. 7b). The resulting network again connects PD-related nodes and exhibits a good overlap with the transcriptional-derived network. This consistent result indicates that the observed transcriptional modification led to an impaired neuronal phenotype, despite the subtle differences

**Table 3 Central nodes of the DEG network are associated with PD (Fig. 6c).**

| Node gene | Role in Parkinson's disease |
|---|---|
| HSPA8 (also known as HSP73, HSC70) | Disaggregation of α-synuclein amyloid fibrils[85] |
| | Autophagy, part of the catabolic pathway for α-synuclein[86] |
| | Mediates mitophagy by regulating the stability of PINK1 protein[87] |
| | Impaired gene expression in sporadic PD[88] |
| EEF1A1 | Mediates activation of heat-shock transcription factor HSF1, prevents α-synuclein aggregation[90] |
| | Interacts with Parkin (PARK2)[82] |
| HNRNPC | Interacts with Parkin (PARK2)[82] |
| | Part of the poly ADP-ribose (PAR) cell death pathway accountable for selective dopaminergic neuronal loss[99] |
| PSMA4 | Part of the Parkinson's disease KEGG pathway[92,93] |
| | Interacts with Parkin (PARK2) and FBX07 (PARK15)[82] |
| CYCS | Role in aggregation of alpha-synuclein[170] |
| | CTD gene-disease associations - Parkinson disease gene set[63] |
| ACTN1 | Interacts with DJ-1 (PARK7)[82] |
| | It is a binding partner of mitochondrial-shaping proteins[171] |
| PGK1 | PGK1 mutation causes vulnerability to parkinsonism[172] |
| | Activation of PGK1 partially restored motor function and slowed disease progression[69] |
| PHB | Regulates dopaminergic cell death in substantia nigra[173] |
| SHH | Play a role in neuroinflammatory response in the MPTP model of Parkinson's disease[174] |
| BRCA2 | Deubiquitinase plays a role in neuronal inflammation[175] |
| VPS39 | It is part of the endocytic membrane trafficking pathway involved in PD and its methylation rates are associated with Parkinson's disease risk[176] |
| | Plays complex functions in endocytic and autophagic pathways[177] |
| UQCRFS1 | KEGG pathway, Parkinson disease[92,93] |
| CNTNAP2 | Differentially expressed in the presence of LRRK2 G2019S mutation, associated with PD[97] |
| | GWASdb SNP-disease associations, Parkinson's disease gene set[63] |
| | Plays a role in the formation of protein aggregates and PD[95,96] |
| CUL3 | Ubiquitin ligase, a potential drug target for Parkinson's disease[84] |
| PLCB4 | GWAS - Parkinson's disease[63] |
| | Motor defect consistent with ataxia in Plcb4-null mice[100] |
| EGLN3 | GEO signatures of differentially expressed genes for diseases—Parkinson's Disease_Substantia Nigra[63] |
| | Prolyl hydroxylase targets substrates for ubiquitination[178] |
| RALGPS2 | Targets include Nurr1, which is associated with Parkinson disease[63] |

Central nodes were determined using the Gephi visualization platform. They represent points of convergence of the network (Supplementary Fig. 5). Since these nodes have already been linked to PD pathways, many more DEGs might also contribute to PD pathology through these pathways. These nodes not only provide a point of convergence for DEGs identified in our study, but they also interact with several PARK genes, suggesting that PARK proteins may also converge on the pathways identified here (Supplementary Fig. 7).

in expression, and further highlights the importance of the proposed PD Core network.

## Discussion

The aim of this study was to identify genes that were differentially expressed as a result of a mutation in the PINK1 gene, using mDA neurons differentiated from patient-derived iPSCs, a model relevant to PD. We focused on the analysis of early timepoints of the differentiation protocol, on cells undergoing neural differentiation up to the state of early postmitotic mDA neurons (D21), as these are not expected to display the activation of damage-control pathways induced by neurotoxicity, but are likely to reveal pathways that lead to primary pathology of PD. Because genetic background can potentially influence the severity and course of the disease[12,83], we chose a cell line homozygous for the ILE368ASN-PINK1 mutation. This mutation imparts a very strong drive towards PD, resulting in full penetrance and an early onset of the disease, hence its impact is expected to diminish the effect of genetic background[12].

By including four different differentiation timepoints and requiring each DEG to be altered at every timepoint, we excluded pathways associated with mDA differentiation, as the expression of such genes changes between each step (Supplementary Fig. 12). The limitation of using an early time period is that we could not identify pathways associated with PD pathology in mature and aging neurons, however, this was intentional. We focused on the identification of pathways prior to damage onset, in order to

eliminate the identification of pathways secondary to the disease, ones induced by damage and associated with cell death. Extension of the timeline to mature and aging neurons is the focus of our future experiments.

Figure 4b shows that samples clustered according to the differentiation stage, rather than cell line identity. Therefore, while the requirement for expression at all timepoints allowed us to identify changes independent of cell state transition, pairwise comparison excluded genes commonly expressed at any particular timepoint, with remaining expression changes being specific to the presence of the PD-associated mutation. The single-cell expression data were analyzed in several layers. First, we identified the most strongly DEGs consistently altered in the same direction at all four timepoints including iPSCs (Fig. 6a and Table 2). This led to a list of genes dysregulated by the PINK1-ILE368ASN mutation independent of the cell type (iPSCs, neuronal precursors, or neurons) (Table 2, Group A "incl. iPSCs", marked by "X"). Second, we applied an approach, in which the iPSC timepoint was excluded, leading to an expanded gene list, which included genes more likely to be dysregulated specifically in a neural cell type (Table 2 and Fig. 6, Group A, 56 genes). Using an approach that reduced the effect of variability between pairs due to different differentiation states expanded the list to 151 genes dysregulated in the same direction at all timepoints (Fig. 6a—Group B and Supplementary Table 4), while taking into account only the absolute degree of change in iPSCs expanded the list further, to 172 non-neuron-specific DEGs (incl. iPSCs, Fig. 6a—group C and Supplementary Table 5). Excluding iPSCs from

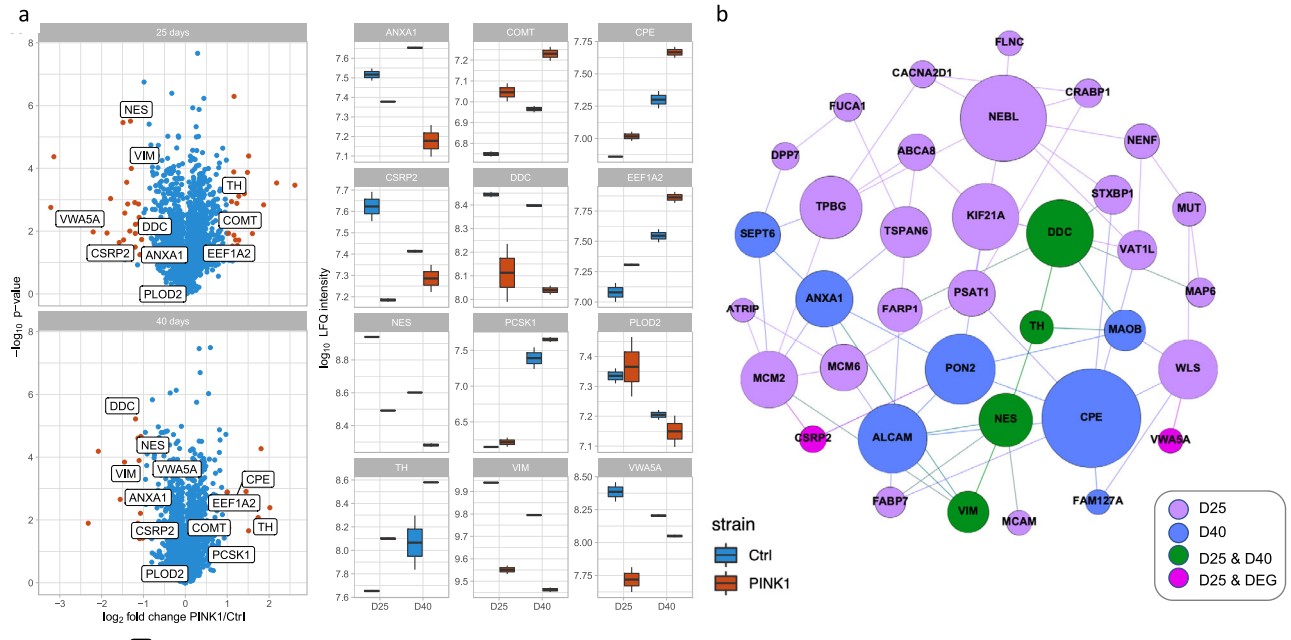

**Fig. 7 Comparative proteomics analysis between CTRL and *PINK1*-ILE368ASN cell line at D25 and D40 validates the manifestation of the transcriptional phenotype.** Results of proteomic analysis at D25 and D40 of the differentiation protocol. **a** The volcano plot shows significantly differentially abundant proteins (FDR <0.05, fold change larger than 2 or −2) as red points, with remaining datapoints shown in blue. The names of proteins that were detected as both top differentially abundant at the protein level by the proteomics analysis and as differentially expressed at the mRNA level by SC-RNAseq are highlighted using a textbox. The data shows results at two timepoints, D25 and D40, in two biological replicates per timepoint. Box plots further highlight the expression of genes shown in textboxes of the volcano plot (interquartile range, showing the expression at D25 and D40, in the PINK1 cell line and in control (IQR, 25–75% q1–q3), with bars indicating Q1 ± 1.5 IQR). **b** This figure shows a network of proteins differentially expressed between a control and a PINK1 mutation-carrying cell line, at D25 and D40. Proteins which are differentially expressed at both D25 and D40 are highlighted in green and point to a dysfunction of the dopaminergic system. D25 differentially abundant proteins are in purple, D40 in blue, proteins also identified as by SC-RNAseq differentially expressed at the mRNA level are in pink. For a table of proteins see Supplementary Table 8. Betweenness centrality was used to illustrate the relative connectedness of each node within the network, the greater the number of documented interactions with other nodes, the larger the circle.

this analysis again expanded this list by neuron-specific DEGs, to a total of 285 (Fig. 6a—group D and Supplementary Data 1). Creating a protein–protein interaction network based on these groups of DEGs demonstrated that genes of all groups formed important nodes within the interaction network. Furthermore, genes of all groups were frequently associated with PD. Overall, this indicates that all the selection criteria levels identified relevant targets (Fig. 6 and Supplementary Data 4).

Analysis of the network shows that certain DEGs are points of convergence within the protein network and form major nodes (Fig. 6 and Supplementary Fig. 9), namely CUL3, HSPA8, EEF1A1, UQCRFS1, CNTNAR2, PSMA4, HNRNPC, and PLCB4. The proteins forming the major nodes are already known to play an important role in PD pathology (Fig. 6c and Table 3). CUL3 has been linked to PD by GWAS studies and is considered a potential PD drug target[84]. HSPA8 (also known as HSP73 and HSC70), disaggregates α-synuclein amyloid fibrils and plays a role in autophagy and the catabolic pathway for α-synuclein, mediates mitophagy by regulating the stability of the PINK1 protein, and its expression was shown to be impaired in sporadic PD[85–88]. In fact, HSPA8 is by far the most important node in the network generated with data from the STRING[59] database, which is preferentially based on functional interaction (Supplementary Fig. 9a, b). It is also one of the most highly dysregulated genes in our dataset. EEF1A1 Translation Elongation Factor mediates activation of the heat-shock transcription factor HSF1, a key player in PD[89], and prevents α-synuclein aggregation, as well as interacting with Parkin (PARK2) and HTRA2 (PARK13)[82,90]

(Supplementary Fig. 9). UQCRFS1 is a mitochondrial electron transport chain ubiquinol-cytochrome c reductase[91], a member of the KEGG-PD pathway (Entry K00411[92,93]), and has been identified as a PD risk gene[94]. CNTNAP2, which belongs to the neurexin superfamily, plays a role in triggering protein aggregates[95,96], was found to be differentially expressed in the blood of PD patients with *LRRK2* mutation[97], and was also associated with PD by GWAS[63]. PSMA4, a proteasome subunit, is part of the KEGG-PD pathway (hsa05012, bta05012, and K02728)[92,93] and is a member of the ubiquitin-proteasomal pathway, which plays a key role in Parkinson's disease[98]. It also interacts with Parkin (PARK2) and FBXO7 (PARK15)[82]. HNRNPC interacts with both PARK2 and members of the Poly (ADP-ribose)-dependent cell death pathway implicated in PD[99]. *PLCB4* has been linked to PD[63] and knock-out mice show motor defects consistent with ataxia[100]. However, many of the less conspicuous nodes are also known to play a role in PD, including EGLN3, IPO5, IPO7, PALLD, PGD, RALGPS2, CYCS, SHH, BRCA2, and others (Fig. 6c and Table 3).

Hence, the network derived from our analysis of the ILE368ASN-*PINK1* mutation is revealing the convergence of many known key PD-associated pathways. This convergence suggests that different mutations may feed into the same PD pathology-associated routes and that each mutation can act through several pathways. A good example of such previously unexplored interaction complexity are the interactions between two prominent PD partners, PINK1 and PARKIN. The PINK1 protein is known to interact with PARKIN directly and together

they target damaged mitochondria for degradation[11–13]. However, our data indicates that the presence of the ILE368ASN-PINK1 mutation results in the dysregulation of several other genes that are possibly upstream of PARKIN[101], including HNRNPC[99], MTRNR2L1[102], MYL12A, and SLC25A4[103], as well as LMAN1, a membrane mannose-binding lectin, which was shown to play a role in PARKIN translocation[104]. This suggests that the direct interaction between PINK1 and PARKIN is not the only means by which PINK1 interacts with the PARKIN pathway.

A strength of our network analysis is that it might shed light on PD-associated genes whose function is so far poorly understood. An example is the mitochondria-localized CHCHD2 protein[105], also called PARK22. Mutations in its gene are linked with autosomal dominant PD, but the precise mechanism is unknown[106]. One hypothesis is that CHCHD2 colocalizes with the mitochondrial contact site and cristae organizing system (MICOS)[106]. However, in the DEG-based protein network, CHCHD2 directly interacts with at least three other proteins, SLC25A4/ANT1 (STRING[59]), GHITM (STRING[59] and GeneMANIA[61]), and NME4 (GeneMANIA[61]). Evidence suggests that GHITM plays a role in PINK1-mediated neurodegeneration[107] and NME4 was shown to be downregulated in PD[75]. SLC25A4 (also known as ANT1) plays an essential role in mitophagy and has been linked to PD pathology[103,108]. Hence, when it comes to mediating pathological changes in CHCHD2-associated PD, the interaction of CHCHD2 with SLC25A4 (ANT1), GHITM, and NME4 may be more relevant than its previously proposed interaction with MICOS in (Fig. 6, in pink).

We also analyzed the correlation of expression between various gene pairs. This correlation may indicate that the genes and their proteins are targets of the same regulatory pathway, or are otherwise related. In our dataset, the expression of several interaction partners shows high correlation, namely PLCB4-RALGPS-TTC3-ZNF37A, EIF3B-HSPA8 (a major network node, ubiquitination pathway)-PCBP1 (ubiquitination pathway). Another cluster centers on MT-CYB and involves both mitochondrial and ubiquitination pathways by NME1–MT-CYB–MT-ND5–MT-CO3–MRPS21 interactions. Among the top pairs are also PSMD7-PSMB5, TAGLN-MYL9, and VWA5A-ZCCHC11 (Supplementary Figs. 16, 17). The interactions of these proteins may, therefore, play a key role in PINK1-mediated PD pathology. To further investigate the involvement of this network in PD, we performed a manual search and found that 68% of the DEGs are already known to be associated with PD (Fig. 7b and Supplementary Data 4), with nearly all major nodes having strong PD association (Table 3 and Fig. 7b). This indicates that these nodes may be key points of integration of the effects of PD pathology, an idea further substantiated by the convergence of the added PARK genes onto these nodes (Supplementary Fig. 9). Furthermore, these nodes form a link between different functional pathways known to be involved in PD. In particular, this is true for CUL3, HSPA8, and PSMA4 (Supplementary Fig. 7 and Supplementary Data 3).

To see whether a reciprocal approach leads to the same conclusion, we looked at whether some of the known PARK proteins directly interact with the network (Supplementary Fig. 9 and Supplementary Table 6). This has revealed that all 19 protein-coding PARK genes form direct interactions with the network, often with several DEGs, as indicated by the size of the node they form when included in the network (Supplementary Fig. 9b, d, DEGs that directly interact with PARK proteins are in pink). Not surprisingly, PARKIN, a known PINK1 partner, was the most strongly associated member of the PARK group with the DEG-based network. The CHCHD2 gene (PARK22) was itself identified as one of the DEGs. The resulting network illustrates that, in spite of the very different nature of PD-associated mutations, the molecular pathways through which the different PARK genes mediate PD pathology are interconnected. In fact, it is often the

central nodes which directly interact with proteins of the PARK genes (Supplementary Table 6), which suggests that PD-associated mutations converge on the same network of central nodes, which then mediate common aspects of PD pathology and would explain why mutations in so many genes lead to a similar outcome[109]. As a corollary, any mutation can lead to pathology via several molecular paths. This allows for the involvement of a network which contains many potential modifiers and underscores the role genetic background plays in PD penetrance and severity, as alleles of several network genes may reduce or amplify the effect of any given mutation[12,83].

Another line of supporting evidence for the network's role in PD is that, based on the STRING[59] database search, the most strongly associated KEGG pathway of this dataset is the Parkinson's disease KEGG pathway (Fig. 5c). CYCS, an important node of the network, is part of the KEGG Parkinson's pathway (Supplementary Fig. 9 and Supplementary Data 5). The other three KEGG pathways identified were spliceosome, Huntington's disease, and thermogenesis, in order of decreasing strength of association (Fig. 5c and Supplementary Data 5).

A surprising finding from our work, which examined neurons during their differentiation and up to their early postmitotic state, is that pathways known to play a key role in PD are profoundly and consistently dysregulated at all timepoints examined, far before the onset of PD pathology. This is in line with current research suggesting that pathology far precedes the onset of notable mDA neuron cell death and observable PD motor symptoms[19,110]. For example, the CHCHD2 protein is part of the purine metabolic pathway that produces DNA, RNA, nucleosides, and nucleotides and has been shown to be altered in PD[75–78]. The DEG network illustrates that the expression of a large number of interconnected genes in the aromatic compound metabolic pathway is altered in cells carrying the PINK1-ILE368ASN mutation (Supplementary Fig. 7b). In total, 39 genes of the nitrogen compound metabolic process (Ncmp) and 88 genes specific to the aromatic compound metabolic process (Acmp, a subgroup of the Ncmp) are differentially expressed in our dataset (Supplementary Data 3). Many of the DEGs identified in our study are part of more than one pathway and, therefore, interconnect the various pathways known to play a role in PD, including stress and catabolic processes[72,73], aromatic compound metabolism[75], vesicle-mediated transport and exocytosis[79], RNA metabolism[74], protein transport, localization, folding, stability, and ubiquitination[70] (Supplementary Fig. 7a–g and Supplementary Data 3). This confirms observations that PD pathology involves many different pathways[111] and suggests that the final stage is a result of long-term untreated pathology. It also points to possible early alterations which may be detectable and used as a diagnostic tool, as well as to targets for early treatment and prevention of the disease.

To investigate whether the observed transcriptional modifications lead to functional deficits that would further support the relevance of this model, we performed a proteomics analysis at D25 and D40 of the protocol. D21 represents early postmitotic neurons, while D25 represents early mature neurons and D40 mature neurons. Our first analysis showed dysregulation of dopaminergic metabolism at D25 and D40 of differentiation (Fig. 7b and Supplementary Table 8). The list of differentially abundant proteins identified by proteomics analysis exhibits an overlap with the DEGs identified in the sc-RNAseq analysis (Fig. 7b and Supplementary Table 8) and many of these proteins are already known to be involved in PD[112]. Importantly, two proteins that were differentially expressed at both D25 and D40, DDC and TH, are key enzymes involved in dopamine metabolism and closely associated with PD[113,114]. Altogether, four proteins were differentially expressed at both timepoints, in two

independent biological replicates per timepoint (Fig. 7b and Supplementary Table 8). The other two proteins are the cytoskeletal proteins VIM (Vimentin)[115] and NES (Nestin), the latter is co-expressed with the PD-associated gene DJ-1 (PARK7)[116]. These were found to be abnormal also by other studies, and are involved in cytoskeletal transport, which represents a key aspect of PD pathology[112]. Performing a network analysis based on the proteome phenotype revealed a proteomics network related to the transcriptional network (Fig. 7b). In all, these data show a consistent abnormality in the levels of enzymes needed for DA metabolism, which indicates that cells carrying the PINK1-ILE368ASN mutation have a functional deficit of the DA metabolic pathway that eventually can lead to neuronal loss of mDA.

The next important step will be to investigate gene expression alterations in aging neurons and how this leads to neurodegeneration in the presence of PD-associated mutations. Genetic background likely plays a greater role in PD caused by mutations with lower penetrance or in idiopathic cases. Therefore, in the future, we will explore the potential overlap between the network identified in this study and the pathways altered by idiopathic disease, as well as the effect of genetic background, by investigating isogenic controls together with cell lines carrying PD mutations. The challenge is that idiopathic cases can be caused by interactions between genes of small effect and the environment, or between environmental factors alone, which potentially broadens the spectrum of pathways involved in the development of PD in these cases[117,118]. Many of these pathways are unlikely to be strongly altered by gene mutations and are likely difficult to distinguish from background noise generated by natural variation in PD-unrelated pathways. Therefore, we first focused to understand the effect of PD-associated mutations of strong effect, in order to detect a core network of pathways distinctly altered in PD.

It will be of great interest to see if cells from idiopathic patients show dysregulation of this integrated network. In fact, our analysis has identified genes, which are known to be associated with sporadic PD, but which had no known connection to molecular mechanisms underlying PD pathology. Knowing how they integrate into the network may point to the mechanism by which they cause PD pathology. For example, one of the top DEGs is LGI1[119]. The development of antibodies to the LGI1 protein leads to immunomodulated Parkinsonism, yet there is no known mechanism linking it to PD pathology[119]. In the network, LGI1 directly interacts with several neighbors (Supplementary Fig. 10). Its most important interaction is its co-expression with CNTNAP2, which is part of the neurexin family and is required for axon organization, and MGMT, which repairs the methylated nucleobase in DNA[59]. From GeneMANIA alone, the strongest evidence is for interaction with GOLT1B, which plays a role in Golgi transport[120]. Hence, LGI1-associated pathology leading to PD symptoms may be mediated through pathways, which are also dysregulated by the presence of the PINK1-ILE368ASN mutation. CNTNAP2 is another very good candidate, as it was shown to be dysregulated in PD patients carrying a mutation in the LRRK2 gene, providing additional evidence that it likely plays a role in PD pathology[97].

The fact that so many genes which belong to other PD mutation-related pathways were dysregulated by the presence of the PINK1-ILE368ASN mutation suggests that pathways involved in PD pathology are far more interconnected than previously thought. It is likely that PD pathology is more a disease with a characteristic network fingerprint than a disease caused by independent mutations acting through unrelated pathways (Fig. 6a). This and future studies will hopefully provide a picture of how various mutations feed into this network and cause its dysregulation. If idiopathic PD is shown to also be mediated by the dysregulation of this network, then we may finally be able to understand the cause of idiopathic PD, which represents 80–85% of all PD cases[2].

## Methods

**Generation of iPSC cell lines.** Fibroblasts (cat. No. ND40066) isolated from a 64-year-old male with PD symptom onset at 33 years of age who carried a homozygous mutation ILE368ASN (P.I368N/P.I368N) (Supplementary Fig. 11) in the PINK1 gene were obtained from the Coriell Institute (Cat. No. ND40066). Samples were collected in accordance with the US Government guidelines and are subject to an MTA issued by Coriell Institute for Medical Research NINDS Cell Repository. Conditions for use of the NINDS Materials are governed by the Rutgers University Institution Review Board (IRB) and must be in compliance with the Office of Human Research Protection (OHRP), Department of Health and Human Services (DHHS), regulations for the protection of human subjects found at 45 CFR Part 46. Patient consent was obtained before collection as per NINDS requirements, described in Supplementary file "NINDS sample submission guidelines & consent" under the section "Sample Submission". Fibroblasts were cultured on gelatin-coated plates (10% gelatin in PBS, coated for 10 min at room temperature) in KO DMEM + 10% FBS + 1% penicillin/streptomycin (stock was 10,000 units penicillin and 10 mg streptomycin ml⁻¹) at standard culture conditions (37 °C, 5% CO2).

Live adherent fibroblasts in culture media were sent to be karyotyped by Cell Line Genetics, Madison, WI, USA (Supplementary Fig. 1) and confirmed to have a normal karyotype. The reprogramming of fibroblasts into pluripotent stem cells was done at Yale Human Embryonic Stem Cell Core (New Haven CT) using the Sendai virus. The iPSC clone was again analyzed using Array Comparative Genomic Hybridization (aCGH), a high-resolution karyotype analysis for the detection of unbalanced structural and numerical chromosomal alterations and confirmed to be normal (Supplementary Fig. 2 and Supplementary Tables 9, 10). To confirm the presence of homozygous PINK1 (P.I368N/P.I368N) mutation, PCR was performed using GoTaq (Promega), Cycling: 95oC 30 s, 36x (95oC 15 s, 60oC 20 s, 68oC 15 s), 68oC 5 min. Primers are listed in Supplementary Table 7 (designed using Primer3Plus and synthesized by Eurogentec). The PCR was confirmed by electrophoresis to produce only one band, the remaining reaction was cleaned using a PCR cleaning kit (Pure Link PCR Micro Kit Cat. 310050). The PCR fragment was sequenced by Eurofins Genomics and sequencing results are listed in Supplementary Fig. 11a, b (the sequence underlying Supplementary Fig. 11 has been deposited to NCBI under the accession OK050183.1). The resulting iPSC cell lines were maintained on Geltrex matrix (Gibco) in mTeSR™1 media (StemCell Technologies) under standard incubator conditions of 5% CO2 and humidity. The protocol was approved by the Committee on Human Research at the University of California San Francisco. The control cell line (also known as 17608/6) is described in ref. [38], it was stained for Oct 3/4 and Tra-1-60 in parallel to the PINK1 cell line. The source is the dermal fibroblasts of a healthy 67-year-old male.

**Analysis of iPSC status and trilineage potential by TaqMan iPSC Scorecard assay.** To confirm the iPSC status of reprogrammed donor fibroblasts, we performed a TaqMan iPSC Scorecard Assay[44], which also confirmed the cells' trilineage potential (Fig. 2b). We followed the protocol described by the manufacturer of the TaqMan hPSC Scorecard Assay (Thermo Fisher Scientific).

Stem cells were cultured on Geltrex matrix (Gibco) in mTeSR™1 media (StemCell Technologies) under standard incubator conditions of 5% CO2 and humidity. On the day of analysis, the cells were dissociated using Accutase and pelleted by centrifugation. RNA was extracted using a Qiagen extraction kit and cDNA was synthesized as per Scorecard kit instructions. Embryonic bodies were generated as per Scorecard kit instructions, RNA was extracted and cDNA synthesized in the same way as for iPSC pellets. The TaqMan hPSC Scorecard Kit 384w plate was amplified using Lightcycler 480 (Roche Diagnostics) and the data were uploaded to the hPSC Scorecard analysis software available online from Thermo Fisher Scientific. The resulting graphs were downloaded and included in Fig. 2.

**Immunocytochemistry.** A 24-well cell culture plate was seeded with iPSCs, one or two wells per cell line, and the IPSCs were then allowed to form colonies. At least a dozen colonies were present in each well and images were taken of several representative stained colonies. This was performed prior to any major experiment, to confirm the status of the cell line. Any evidence of differentiation identified by a loss of iPSC marker expression was documented. These adherent colonies were fixed in 4% PFA for 10 min, washed and permeabilized with 0.1% Triton X-100 in 1X PBS for 15 min, then washed and incubated in a blocking solution of 2% BSA in 1X PBS for 1 h. They were then incubated with a primary antibody for POU5F1 (also known as Oct 3/4, Santa Cruz Biotechnology, sc-5279) and TRA-1-60 (MAB4360, Merck Millipore) at 1/500 dilution in blocking solution, overnight at 4 °C (Fig. 2a). The next day they were washed three times with PBS and a secondary antibody (AlexaFluor 488, Thermo Fisher) was applied at a 1/1000 dilution in blocking solution and incubated for 1 h at room temperature. The cells were then washed three times with PBS and imaged. Differentiated cells were stained for microtubule-associated protein 2 (MAP2, MAB3418, Merck Millipore), tyrosine hydroxylase (TH, Pel-Freez Biologicals P40101), PAX6 (901301, Imtec diagnostics) at 1/500 dilution, PITX3 (Sigma-Aldrich, HPA044639), LMX1A (Abcam,

ab139726) and SLC6A3/DAT (Thermo fisher, PA1-4656). (Supplementary Table 7 and Supplementary Figs. 3, 13, 14). Images were captured using a confocal Zeiss Laser Scanning Microscope 710 with a 20x air objective and processed using ZEISS ZEN Microscope Software. The same preset parameters were used for the acquisition of images. Images were converted from .czi format to .tiff format and scale bars were added using Fiji open-source software[121].

**Differentiation of iPSCs into mDA neurons.** The protocol used to differentiate iPSCs into mDA neurons was modified from refs. [24,122] (Table 1). The iPSCs were grown to 95% confluence, dissociated using accutase, and 1.5 wells were combined into one well at day −1. They were allowed to recover in the presence of ROCK inhibitor for about 8 h and then in mTeSR without ROCK inhibitor for about 16 h. After this, day 0 media were applied (Table 1). Both control and PINK1-ILE368ASN cell lines were differentiated at the same time so that they would be subject to the same conditions. Different timepoints were generated by repeating the differentiation protocol on a later date, as described in Supplementary Table 2.

Cells were fed fresh media daily, 36 ml per six-well plate or as needed, judging consumption from media color, and replacing media whenever it started to turn yellow, using the appropriate media and factor mix for that day.

**Real-time quantitative PCR of mDA and non-mDA markers.** Total RNA was extracted from a cell pellet of a 12-well plate well using the RNeasy Plus Universal Kit Mini (50), Catalog no. 73404), as per manufacturer instructions. The RNA concentration was determined through absorption at 260 nm using the Nanodrop instrument (Fisher Scientific). The Superscript IIITM First-Strand Synthesis System for RT-PCR (Invitrogen) was used to prepare cDNA, using oligo(dT)20 and 2 ug of total RNA as per manufacturer instructions. The cDNA was stored at −20 °C.

Primers were designed using Primer Blast[123] and synthesized by Eurogentec Belgium. The primers used are listed in Supplementary Table 7. Standard templates of 90–150 bp in length were generated by PCR, purified using Invitrogen Pure Link PCR Micro Kit (K310050), and their concentration determined using NanoDrop Spectrophotometer. These were then diluted to generate a series of standards of known concentration, from 200 to 0.002 fg µl⁻¹. The cDNA levels within samples were determined using quantitative real-time PCR (QRT-PCR) on a Roche Lightcycler 480 using the Maxima® SYBR Green/ROX qPCR Master Mix (2×) (cat. #K0223) using absolute quantitation by generating a standard curve based on the standards of known concentration and extrapolating the concentration of the unknowns (samples). The parameters were: initial denaturation at 95 ºC for 10 min., followed by 40 cycles of 95 ºC for 15 s, 60 ºC for 30 s, and 72 ºC for 35 s. This was followed by a dissociation curve to confirm that only one PCR product was present. Each absolute concentration of a particular gene was then divided by the absolute concentration of a housekeeping gene, in this case, GAPDH. In previous experiments, GAPDH has been identified as the most stable housekeeping gene in iPSCs and in iPSCs differentiating using our protocol. The values were, therefore, standardized per total RNA of the sample, since 2 ug of total RNA was used for every sample, as well as per expression GAPDH.

**Statistics and reproducibility.** In real-time qPCR graphs, each timepoint consists of at least three independently differentiated samples, seeded at the same time, hence representing biological replicates. The sample concentration was determined by absolute quantitation, comparing the sample concentration to a known concentration of a standard template identical to the one being amplified. The value was standardized to total RNA, by cDNA synthesizing each cDNA sample from a standard amount of total RNA for each sample. This value was then divided by the concentration of GAPDH obtained for that particular sample, thus standardizing to GAPDH levels and generating a unitless number denoting expression relative to the expression of the housekeeping gene GAPDH. GAPDH was selected from among a number of possible housekeeping genes, as it showed the best ability to normalize gene expression in a population of untreated samples. A detailed description of this rationale and approach is in Novak et al.[124]. Each of the samples was amplified in duplicate. Each sample value was an average of the experimental duplicate. Standard error was calculated as the standard deviation of the three biological replicates, divided by the square root of the number of samples[125].

To allow for reproducibility through independent analysis, all datasets were made accessible and can be accessed from repositories listed in the Code availability and Data availability sections.

**Single-cell RNA sequencing.** On the day of collection, cells were dissociated using accutase. The single-cell suspension was spun down and cells were washed with (PBS, 2% BSA) twice, then passed through a 40 µm filter to remove larger cell clumps. The sample was then counted and viability was determined using (Vi-CELL XR, Cell Counter, Beckman Coulter). Cells were required to have at least a 95% viability. The samples were then diluted in PBS with 2% BSA to a final concentration of 190,000 cells ml⁻¹. About 3 ml were then used for single-cell analysis. Subsequently, cells were processed by the Drop-Seq approach[37,126,127] and sequenced.

**Microfluidics fabrication for single-cell RNAseq.** Microfluidics devices were generated on-site, using a technique described below, which is based on an earlier Drop-Seq protocol[37,128,129]. Soft lithography was performed using SU-8 2050 photoresist (MicroChem) on a 4″ silicon substrate, to generate a 90 µm aspect depth feature. The wafer masks were subjected to silanization overnight using chlorotrimethylsilane (Sigma), before being used for the fabrication of microfluidics. Silicon-based polymerization chemistry was used to fabricate the Drop-Seq chips. In short, we prepared a 1:10 ration mix of polydimethylsiloxane (PDMS) base and cross-linker (Dow Corning), which was degassed and poured onto the Drop-Seq master template. PDMS was cured on the master template, at 70 °C for 2 h. After cooling, PDMS monoliths were cut and 1.25 mm biopsy punchers (World Precision Instruments) were used to punch out the inlet/outlet ports. Using a Harrick plasma cleaner, the PDMS monolith was then plasma bonded to a clean microscope glass slide. After the pairing of the PDMS monolith's plasma-treated surfaces with the glass slide, we subjected the flow channels to a hydrophobicity treatment using 1H,1H,2H,2H-perfluorodecyltri-chlorosilane (in 2% v/v in FC-40 oil; Alfa Aesar/Sigma) for 5 min of treatment. Excess silane was removed by being blown through the inlet/outlet ports. Chips were then incubated at 80 °C for 15 min.

**Single-cell isolation and RNA capturing.** We determined experimentally that, when using the microfluidics chips, a bead concentration of 180 beads/µL is optimal for an efficient co-encapsulation of the synthesized barcoded beads (ChemGenes Corp., USA) and cells, inside droplets containing lysis reagents in Drop-Seq lysis buffer medium. Barcoded oligo (dT) handles synthesized on the surface of the beads were used to capture cellular mRNA.

For cell encapsulation, we loaded into one syringe each, 1.5 ml of bead suspensions (BD) and the cell suspension. Micro-stirrer was used (VP Scientific) to keep beads in homogenous suspension. For the droplet generation, a QX200 carrier oil (Bio-Rad) was loaded into a 20-ml syringe and used as a continuous phase. To create droplets, we used KD Scientific Legato Syringe Pumps to generate 2.5 and 11 ml/h flowrates for the dispersed and continuous phase flows, respectively. After the droplet formation was optimal and stable, the droplet suspension was collected into a 50-ml Falcon tube. In total, 1 ml of the single-cell suspension was collected. Bright-field microscopy using INCYTO C-Chip Disposable Hemacytometer (Thermo Fisher Scientific) was used to evaluate droplet consistency and stability. To avoid multiple beads per droplet, bead formation and occupancy within individual droplets was monitored throughout the collection process.

The subsequent steps of droplet breakage, bead harvesting, reverse transcription, and exonuclease treatment were carried out as described below, in accordance with the Drop-Seq protocol[37]. The RT buffer was premixed as follows, 1× Maxima RT buffer, 4% Ficoll PM-400 (Sigma), 1 µM dNTPs (Thermo Fisher Scientific), 1 U/ml RNase Inhibitor (Lucigen), 2.5 µM Template Switch Oligo, and 10 U/ml Maxima H-RT (Thermo Fisher Scientific). After Exo-I treatment, INCYTO C-Chip Disposable Hemacytometer was used to estimate the bead counts, and 10,000 beads were aliquoted in 0.2 ml Eppendorf PCR tubes. We then added 50 µl of PCR mix, consisting of 1× HiFi HotStart ReadyMix (Kapa Biosystems) and a 0.8 mM Template Switch PCR primer. The thermocycling program of the PCR was 95 °C (3 min), four cycles of 98 °C (20 s), 65 °C (45 s), 72 °C (3 min) and 9 cycles of 98 °C (20 s), 67 °C (20 s), 72 °C (3 min), and a final extension step of 72 °C for 5 min. After PCR amplification, 0.6× Agencourt AMPure XP beads (Beckman Coulter) were used for library purification according to the manufacturer's protocol. The purified libraries were eluted in 10 µl RNase/DNase-free Molecular Grade Water. We used the Bioanalyzer High Sensitivity Chip (Agilent Technologies) to analyze the quality and concentration of the sequencing libraries.

**NGS preparation for Drop-seq libraries.** The 3′ end-enriched cDNA libraries were prepared by tagmentation reaction of 600 pg cDNA library using the standard Nextera XT tagmentation kit (Illumina). Reactions were performed according to the manufacturer's instructions. The PCR amplification cycling program used was 95 °C 30 s, and 12 cycles of 95 °C (10 s), 55 °C (30 s), and 72 °C (30 s), followed by a final extension step of 72 °C (5 min). Libraries were purified twice to reduce primers and short DNA fragments with 0.6× and 1× Agencourt AMPure XP beads (Beckman Coulter), respectively, in accordance with the manufacturer's protocol. Finally, purified libraries were eluted in 10 µl Molecular Grade Water. Quality and quantity of the tagmented cDNA library were evaluated using Bioanalyzer High Sensitivity DNA Chip. The average size of the tagmented libraries prior to sequencing was between 400 and 700 bps.

Purified Drop-seq cDNA libraries were sequenced using Illumina NextSeq 500 with the recommended sequencing protocol except for 6 pM of custom primer (GCCTGTCCGCGGAAGCAGTGGTATCAACGCAGAGTAC) applied for priming of read 1. Paired-end sequencing of 20 bases (covering the 1–12 bases of random cell barcode and the remaining 13–20 bases of random unique molecular identifier (UMI)) was performed for read 1, and of 50 bases of the genes for read 2 (Supplementary Fig. 4).

**Bioinformatics processing and data analysis.** The FASTQ files were assembled from the raw BCL files using Illumina's bcl2fastq converter and run through the

FASTQC codes (Babraham bioinformatics; https://www.bioinformatics.babraham.ac.uk/projects/fastqc/) to check for consistency in library qualities. The monitored quality assessment parameters monitored were (i) per-base sequence quality (especially for the read 2 of the gene), (ii) per-base N content, (iii) per-base sequence content, and (iv) over-represented sequences. The FASTQ files were then merged and converted into binaries using PICARD's FastqToSam algorithm. The sequencing reads were converted into a digital gene expression matrix using the Drop-seq bioinformatics pipeline[37].

**Single-cell RNAseq data analysis**. The identification of low-quality cells was done separately for each dataset. In order to select only the highest quality data, we sorted the cells by their cumulative gene expression. Only cells with the highest cumulative expression were considered for the analysis[130].

In addition to this filtering, we defined cells as low-quality based on three criteria for each cell. The number of expressed genes must be more than 200 and 2 median absolute- deviations

(MADs) above the median; the total number of counts has to be 2 MADs above or below the median, and the percentage of counts to mitochondrial genes has to be 1.5 MADs above the median. Cells failing at least one criteria were considered as low-quality cells and filtered out from the further analysis. Similar to the cell filtering, we filtered out low-quality genes, identified by being expressed in less than ten cells in the data.

The integration of the filtered matrices of the different datasets was performed using scTransform[131] on a Seurat object[132] based on the treatment. The final gene expression matrix, which was used for the downstream analysis, consisted of 4495 cells and 39,194 genes with a median total number of mRNA counts of 7750 and a median number of expressed genes of 3521. Principal component analysis (PCA) was computed using the 5000 most variable genes of the integrated data. The clustering of data were performed using Louvain clustering. The resolution of the clustering was selected based on the best silhouette score of the different resolutions[133]. A shortlist of manually curated markers was used to validate the different stages of the differentiation process.

We then performed differential expression analysis between the two treatments (control and PINK1) at each timepoint. The differential expression analysis was done using MAST[132] (default parameters) on the normalized counts using the total number of transcripts in each cell as a covariate and the Bonferroni correction to correct for multiple hypothesis testing (Padj). In addition, we tried to find conserved markers among the different timepoints using MAST again and the total number of transcripts in each cell as a latent variable. Genes with fold changes of the same sign in the fold change were then identified across the different timepoints and the average fold change was calculated. The genes with average fold change > 0.1 and maximum adjusted $p$ value < 0.01 were selected as differentially expressed.

The first analysis of pairwise differential expression at each timepoint (adjusted $p$ values ($p_{adj}$) <0.01 fold change (FC) >0.1) was performed to identify genes that were upregulated and downregulated in the PINK1 cell line compared to control (see Results section). The analysis was repeated with the exclusion of iPSCs and using only D6, D15, and D21 timepoints. We then used the maximum adjusted $p$ value in a pairwise combination as an adjusted $p$ value, and the average fold change that occurred in the pairwise comparison as fold change threshold hence retained only genes dysregulated in the same direction at all timepoints (Group B). We then took the mean of FC of the different timepoints to reduce the effect of the variability between pairs due to their different differentiation states. The analysis was performed for the four timepoints (iPSCs, D6, D15, and D21), taking into account only the absolute degree of change in iPSCs (Group C). The analysis was then repeated using only timepoints D6, D15, and D21 (Group D).

**Network analysis**. We extracted protein–protein interaction information between the DEGs from STRING[59] and from GeneMANIA[61]. We excluded indirect association, such as text mining, co-occurrence, and neighborhood from STRING, and co-expression, colocalization, shared protein domains, and predicted interactions from GeneMANIA, retaining only genetic interactions, pathways, and physical interactions (2122 interactions in total). We deleted any genes or interactions that were added by these databases, in order to only focus on DEGs and interactions among them. The network diameter was calculated and betweenness centrality was used to illustrate the relative importance of each node within the network. As a control, we selected the same number of genes at random, using the list of genes detected by our RNAseq analysis, excluding DEGs. This control set did not produce a network and led to a mostly disconnected array of genes (Supplementary Fig. 5). Networks were also generated using the STRING and GeneMANIA inputs independently (Supplementary Fig. 9). We constructed a correlation network based on the correlation of expression of DEGs ($p$ value <0.05, correlation >0.1) and identify edges that are common to the two networks. This network consisted of 860 interactions (Supplementary Fig. 16). We then extracted shared interactions of these two networks, which amounted to 297 interactions (Supplementary Fig. 17a).

In order to validate the PPI network produced by STIRNGdb (v10), we created 50 PPI (protein–protein interaction) networks using 292 random genes (same as the number of DEGs). We then compared the number of detected proteins, the number of interactions between the genes, and the distribution of the node degrees. We performed the Wilcoxon test to access if the two-degree distributions are

different from one another in a statistically significant manner, which showed a statistically significant difference ($p = 2.22e-16$) (Supplementary Fig. 17b).

**Proteome analysis**. Cell pellets were lysed in 1% sodium deoxycholate in 50 mM sodium bicarbonate pH8. After sonication, samples were incubated on ice for 30 min and centrifuged at 4 °C for 30 min at 16,000×g. Supernatants were recovered and quantified using PierceTM BCA Protein Assay Kit (23225, Thermo Scientific). Protein extracts (10 µg) were reduced with 10 mM DTT for 45 min at 37 °C, incubated for 15 min at room temperature, then alkylated with 25 mM iodoacetamide for 30 min at room temperature in darkness. Proteins were further digested overnight at 37 °C with 0.2 µg of trypsin/Lys-C Mix (V507A, Promega). Samples were acidified in 1% formic acid and centrifuged for 10 min at 12,000 × g. Supernatants were recovered and peptides were desalted on Sep-Pak tC18 µElution Plates (Waters, 186002318), dried by vacuum centrifugation, and reconstituted in 25 µl of 1% Acetonitrile/0.05% trifluoroacetic acid.

Following quantification by nanodrop, each sample (200 ng) was analysed by mass spectrometry. The LCMS setup consisted of a Dionex Ultimate 3000 RSLC chromatography system configured in column switching mode. The mobile phases A and B consisted of 0.1% formic acid in water and 0.1% formic acid in acetonitrile, respectively. The loading phase consisted of 0.05% trifluoroacetic acid and 1% acetonitrile in water. The LC system was operated with a Thermo pepmap100 C18 (2 µm particles) 75 µm × 15 cm analytical column (loading 5 ul min$^{-1}$; analytical 300 nl min$^{-1}$). The loading column consisted of Thermo pepmap100 C18 (3 µm particles) 75 µm × 2 cm. Samples were separated by a linear gradient ranging from 2% B to 35% B 66 min and sprayed into the mass spectrometer using a Nanospray Flex (Thermo Scientific) ion source. MS acquisition was performed on Q Exactive-HF (Thermo Scientific) operated in data-dependent acquisition mode. MS cycle (AGC MS1 3e6; AGC MS2 1e5) consisted of a high-resolution survey scan (60,000 at 200 m/z) followed by the fragmentation of the top 12 most intense peptides at a resolution of 15,000 at 200 m/z. Dynamic exclusion of already fragmented peptide ions was set to 20 s.

Analysis was performed with the MaxQuant software package version 1.6.17.0[134]. The minimum ratio for LFQ was set to 2. An FDR <1% was applied for peptides and proteins. A human Uniprot database (July 2018) was used to perform the Andromeda search[135]. Oxidized methionine and acetylated N-termini were set as variable modifications while carbamidomethylation on cysteine was set as a fixed modification. Peptide tolerance was 20 ppm. MS intensities were normalized by the MaxLFQ algorithm[136] implemented in MaxQuant while using the match-between-runs feature.

**Ethics**. Patient-derived cell lines were handled according to the ethics guidelines set out by the National Ethics Board of Luxembourg, (Comité National d'Ethique dans la Recherche; CNER). The use of these cell lines is governed by a materials transfer agreement (MTA) with the NINDS (fibroblast supplier), which states that the conditions for use of the NINDS Materials are governed by the Rutgers University Institution Review Board (IRB) and must be in compliance with the Office of Human Research Protection (OHRP), Department of Health and Human Services (DHHS), regulations for the protection of human subjects found at 45 CFR Part 46. Patient consent was obtained before collection as per NINDS requirements. Samples are collected with informed consent (under IRB approval) and the process is described in Supplementary file "NINDS sample submission guidelines & consent" under the section "Sample Submission". https://catalog.coriell.org/1/NINDS/About/NINDS-Repository-FAQ; https://stemcells.nindsgenetics.org/?line=ND40066

**Reporting Summary**. Further information on research design is available in the Nature Research Reporting Summary linked to this article.

## Data availability
The sequence underlying Supplementary Fig. 11 has been deposited to NCBI under the accession OK050183.1. Single-cell RNAseq data is available through the Gene Expression Omnibus (GEO), accession number GSE183248. The proteomics data is available via the Proteomics Identification Database (PRIDE), identifier PXD028283. The proteomics dataset is available at https://r3lab.uni.lu/frozen/cca2-s098, with a https://doi.org/10.17881/cca2-s098.

## Code availability
All analysis scripts are publicly available via: https://gitlab.lcsb.uni.lu/ICS-lcsb/ipscs_pink1.

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

## Acknowledgements

We thank Dr. Rudi Balling for stimulating discussions. We would like to acknowledge the excellent support by the LCSB Bio-Imaging Facility through Aymeric d'Hérouël, his advice was essential for the analytical part of this manuscript, as well as for the imaging sections, and the outstanding service by the LCSB-Sequencing Platform through Rashi Halder. We thank Francoise Chanut for her very thorough and effective feedback. D.K. and M.B. were financially supported by the PRIDE program of the Luxembourg National Research Fund through PRIDE17/12244779/PARK-QC and PRIDE/10907093/CRITICS, respectively. This work was also made possible in part by the support of S.F. from the Michael J Fox Foundation through Head Start Program and Parkin Consortium grants and from the National Institutes on Aging (RF1 AG058476 and P01 AG54407).

## Author contributions

G.N. and A.S. designed the project. G.N. generated the iPS cell lines and performed iPSC characterization, as well as mDA differentiation, sample preparation for single-cell RNAseq, qPCR experiments, and ICC. K.G. performed scRNAseq experiments. D.K. designed the bioinformatic analysis pipeline, performed the bioinformatic analysis, and rendered figures. M.B. artistically rendered article figures, assisted with cell culture, and supplied PAX6 staining figures. G.N. designed and performed the network analysis. G.N. and A.S. supervised the work and wrote the manuscript. The iPSC cell lines were generated under the supervision of S.F., who also helped edit the manuscript. G.D. and S.R. performed proteome analysis. G.N. performed interpretation of the proteomics data.

## Competing interests

The authors declare no competing interests.
