## [Peer Review File · Communications Biology]

Reviewers' comments:

Reviewer #1 (Remarks to the Author):

The manuscript by Novak et al. entitled "Single cell transcriptomics of human PINK1 iPSC differentiation dynamics reveal a core network of Parkinson's disease" report the analysis of single cell gene expression profiles of dopaminergic neurons generated from iPSCs that were derived from a patient with the ILE368ASN mutation in PINK1 gene. The main conclusion is that several PARK genes may mediate PD pathology through a few central pathways network, which corroborate previous reports. The manuscript is well written and gene expression analysis was systematically performed. However, there are some controls lacking in the analysis that will strengthen the conclusions:

- 1) Although the author highlight the importance of genetic background in the developing Parkinson's disease the controls used in this study didn't take this in consideration. For instance, the gene editing of the PINK1 mutated iPSC line to generate an isogenic control would be an appropriate control for rigorous differential gene expression analysis strengthening the conclusions.
- 2) Another important control lacking is the use of iPSC lines from idiopathic cases, this would be important to integrate into the analysis to confirm or refute some of the common PD mechanisms proposed. With such comparison one may infer a core network of PD proposed.
- 3) Data/photos showing the quality and the efficiency of the differentiation process into mDA neurons were lacking. Photos showing the differentiation / morphologies and efficiency of the mDA neuron differentiation and the quantitative analysis of the time course differentiation would be important to add.
- 4) Adding in vitro functional studies demonstrating the function of critical gene(s) in the iPSC model reported will further strengthen the conclusions.

Reviewer #2 (Remarks to the Author):

Overview

The publication "Single cell transcriptomics of human PINK1 iPSC differentiation dynamics reveal a core network of Parkinson's disease" provide new insights into ILE368ASN PINK1. The overall results are interesting, however the lack of information about the ILE368ASN mutant in the introduction as a justification to study, the vagueness of the statistical approaches used in the data, the questionable adjustment of pvalue, low fold change cutoff and gene directionality constraints, makes a lot of the interesting data questionable. I think with some fundamental changes to both how this paper is written and the statistical analysis, you could have a great paper defining ILE368ASN PINK1.

Major

Page 6 – Did you gain control from ref28 and run the experiment with these controls or did use the control results from ref28? If it's the later, I will have further comments about the statistical analysis. From my reading of ref28 am I correct in thinking that you used the control cell lines in your experiment and performed the iPSC experiment in conjunction with the PIK1 mutants?

Figures – Text in all figures should be legible, this includes all figures. It is difficult to determine the results when I am unable to even read the text on the figure. For example, figure 2b, between the resolution of the figure and the small text, none of this data can be interpreted.

Statistical tests and numbers are not clearly defined in the results or the figure legends. This makes interpretation of the data difficult as trying to determine if gene sets are previously know in the literature or derived from the analysis.

Figure 5c D21 volcano plot does not look correct. This may be because of the maximum adjusted p-value statistic, over traditional methods, or this may be due to the gene directional change constraint that you imposed on the analysis, which is not appropriate for iPSC stage analysis as gene expression fluctuate over the course of differentiation and so might the gene involved in PINK1.

Network analysis – Correlation of gene expression changes and protein level changes is low, which is further exacerbated by the low fold change cut off > 0.1 . This is further emphasized by the statistical model gaining the list of differentially expressed genes. This means interpreting the network analysis is difficult. Rather than protein-protein interaction networks what did the traditional pathway network analysis show? You state an interaction with known PD genes, I am curious as to if you performed a KEGG analysis, is PD pathway the most highly associated pathway?

Minor Introduction

Page 4 – “clearly plays a role in mitophagy”, “clearly” is not the correct language here, please state that multiple publications have shown PINK1s’ involvement in mitophagy. This is important as your next sentence states it has not been fully elucidated.

Page 4 – “impossible to study them”, this is a highly controversial statement, as publications using microarray and even laser dissected RNAseq has been able and published to show you can study these. Please change the tone of this sentence to state that it has been difficult to study these neurons.

Page 4 – This comment continues as research was not only limited to animal model, please change.

Page 5 – Rather than stating “hence their study would not be as pertinent”, please state that using your model of mDA allows for the elucidation of mechanisms causing PD-induced cell death.

Page 5 – “which in turn leads to the classic movement symptoms”, which it turn has been directly associated to the classic movement symptoms.

Introduction – Could you please write one or two sentences about the mutation ILE368ASN, and any previous knowledge we know about this specific mutation. A short summary of the literature to better understand why you chose this mutation. Readers need to understand the logic of your decision to study this particular mutation. Is PINK1 even expressed in mDA, because you should mention this.

Minor Results

Results – Please provide a sample identification supplementary table with sample information and timepoints used in the experiment. I may have missed it, but it would make reading this publication a lot easier. This would also help with which covariates you used in your analysis.

Page 7 – “iPSC status was ascertained by standard methods”, please provide reference for this statement. “TaqMan iPSC Scorecard Assay”, again provide reference. None of these clearly described in the methods. Please update methods section.

Page 7 – Figure 3a Where did the gene to confirm characterization of iPSCs expressed genes come from? Do you have a reference? Or are these differentially expressed genes between the timepoints of the controls? If they are why do you not state, the model you used? This is also not stated in the figure legend.

Page 8 – This part was well written and concise. The only change is to clearly state the model, are you comparing only controls across differentiation states? Are you comparing all samples and regressing out the PINK1 effects in the model? Or more likely is this just average expression across all the controls?

Page 8 – It is surprising that you only identified so few differentially expressed genes. Was there

unwanted variation that could account for this? Did you try any unwanted variation removal? Were there any multicollinearity issues with the data? With such a complicated question, did you consider linear mixed effect models to account for the time component?

Page 9:10 – I need literature justification of a foldchange of > 0.1 being biological important. With the low number of differentially expressed genes, using a maximum adjusted p-value may not be appropriate as genes are going to change between iPSC stages. Limiting the analysis to genes that have to change in the same direction is not biologically sound as iPSC conversion is not a linear gene expression change. Genes fluctuate across stages and so would disease genes. This also may explain the strange volcano plot of D21 in Figure 5c.

Data integration reveals a common PD network – With the questions about the statistical tests, maximum adjusted p-value and the gene direction constraints I will not go into too much detail about the PPI. I believe the data and the overall PPI results are correct, and this part is well written. The constraints to gene direction and statistics should be reassessed. I think doing a simple traditional pathway analysis, using ranked based gene expressed (GSEA), would alleviate a lot of the questions I have about how you came to the list of differentially expressed genes.

Minor Methods

Statistics – Title should be qPCR statistics, you should also mention the actual statistics used to determine significance and you should mention which method you used (delta delta ct) and you need to mention how you normalized to housekeeping genes (name them) and how many housekeeping genes you used.

Single-cell RNA-seq data analysis – Please provide a reference to the quality filtering parameters. These should be the same as the standard for Dropseq and please provide reasoning for any that are not standard in the literature.

Single-cell RNA-seq data analysis – For the MAST analysis did you use a contrast matrix approach to weight the variability of the entire experiment, if not why? Also, which covariates did you adjust for in your model?

Single-cell RNA-seq data analysis – Fold changes > 0.1 is not a standard in the literature, can you please provide a reference where this has been implemented.

Data availability – For constancy in bioinformatics approaches, many high impact journals now require github repositories for all code used the analysis. This is important to both clarity in analysis and reproducibility, furthermore I would not need to ask basic statistical questions in my review process such as contrast matrices. Are you going to make your code available and if not justify your reason?

Minor Figures

Figure 1. – Place numbers of replicates in figure legend

Figure 2. – Text larger, current figure text is illegible. What stage of differentiation done was the IHC? What was the sample size?

Figure 3. – Text larger, current figure text is illegible. Again, provide n number for experiments and the statistical tests. Is heatmap a unsupervised clustered? Is this differentially expressed genes or a predetermined list from a publication in 3a? You also need to put models in either the text or the figure legend to make this clear. Also, for the qPCR, what housekeeping genes did you use, what statistical test. All these things should be clearly stated.

Figure 4. – Text larger, however this figure is just legible when zoomed in. I would also like to know if you only used controls for this analysis.

Figure 5. – Text larger, statistical models and tests need to be supplied in the figure legend. Also the sample numbers need to be supplied. Figure 5c volcano plot need to be reassessed.

Figure 6. – Completely illegible.

Reviewer #3 (Remarks to the Author):

This manuscript combines single cell transcriptomic approaches with the generation of midbrain dopaminergic neurons obtained from human induced pluripotent stem (iPSc) from patients with Parkinson's disease (PD), to report unified networks underlying sporadic and mutations driven pathology.

The manuscript is organized and well written. This research will provide a great resource for other groups exploring the next steps for treatment and characterization of PD. It was a pleasure to read this important manuscript.

At the same time, my only major comment is that the data presented in this manuscript are largely descriptive. Although substantial computational predictions were provided, direct impact of DEGs on cellular state and neuronal properties was not further tested. Causality of the interactions needs to be demonstrated. Moreover, whether abnormal gene expression impairs key PD-related mechanisms or pathways, including mitochondrial, protein processing or RNA metabolism dysfunction, need to be assessed and demonstrate.

I offer below a list of suggestions and comments

1. What are the evidences (beside abnormal gene expression) that these iPSCs derived mDA neurons are indeed recapitulating some of the PD pathophenotype? The authors could utilize their model and demonstrate by IHC or RNA in situ hybridization, temporal and spatial expression of these genes. More importantly, it will be interesting to reveal new abnormal neuronal properties (early onset) that arise prior to neuronal death (i.e. oxidative stress, DNA breaks, mitochondrial dysfunction, mislocalization of protein)

2. In the discussion, the authors offer a list of genes (such as LGI1, CHCHD2) which "had no known connection to molecular mechanisms underlying PD pathology, though some of them are known to be involved in sporadic PD". Since I think this is one of their major findings, the authors should pick one or two of these pathways and further demonstrate the causality of interactions with PD-related mechanisms. For example, LGI1-CNTNAP2 interaction could be further validated by examination of axon organization.

Minor comments:

- Network figures (Fig 6. and Supplementary Figure 11.) are impossible to see, please revise
- Pg. 17 Line 365, "though some of them are known"

Dear Reviewers,

We would like to thank you for your careful and very valuable assessment of our manuscript. Based on your suggestions, we made it our priority to address the major revisions.

In particular, we agree that **major revisions** should include:

- (1) Additional data to validate the dopaminergic differentiation model (noted by Reviewer 1)
- (2) Analysis of potential neuronal deficits in PD patient-derived iPSCs
- (3) A full rebuttal/revision of statistical and pathway analyses, as noted by Reviewer 2

We performed substantial modifications to the manuscript in order to address the raised questions. We hope that the following experiments have addressed these concerns.

(1) We validated the dopaminergic differentiation model and extended the differentiation trajectory analysis by using additional timepoints available for our control cell line. This was performed by the analysis of mDA marker expression during the differentiation protocol (via qPCR), as well as by staining for key mDA markers, including PITX3, LIMX1A, TH and DAT at day 25 and 35 of the differentiation protocol (Fig. 3 & 4). Furthermore, we used the mDA differentiation gene expression profile recently published by Ásgrímsdóttir and Arenas (2020)¹ to show the progression from radial glia to progenitors to early and then mature mDA neurons within our samples. This characterization is shown in the newly added Figure 3, which focuses on describing the mDA in-vivo differentiation process.

(2) In order to investigate the possible presence of neuronal deficits, we initiated a collaboration with the Dittmar lab at the Luxembourg Institute of Health (LIH) and performed a systematic proteomics analysis at two time points (day 25 and day 40). The results have not only validated the manifestation of key differentially expressed genes on the protein level, but subsequent differential protein abundance based network analysis has also confirmed dopaminergic metabolism deficits. These results are summarized in the newly added Figure 7 as a validation approach. Due to their contribution, Prof. Dr. Gunnar Dittmar and his post-doc Dr. Sophie Rodius from the LIH are now co-authors on the manuscript.

(3) We revisited and partly adapted the statistical and pathway analyses based on the comments of Reviewer 2, which has further strengthened our general conclusion.

Please find below our detailed point-by-point replies to the reviewers' comments.

Reviewers' comments:

Reviewer #1.

The manuscript is well written and gene expression analysis was systematically performed. However, there are some controls lacking in the analysis that will strengthen the conclusions:

1) Although the author highlight the importance of genetic background in the developing Parkinson's disease the controls used in this study didn't take this in consideration. For instance, the gene editing of the PINK1 mutated iPSC line to generate an isogenic control would be an appropriate control for rigorous differential gene expression analysis strengthening the conclusions.

Answer: Dear reviewer, thank you for the comment. We very much agree that the genetic background effect is an important factor to consider. To explain our approach, the following passages were added to the Discussion section (lines 356-360 and 513-518).

“Because genetic background can potentially influence the severity and course of the disease^{2,3}, we chose a cell line homozygous for the ILE368ASN PINK1 mutation. This mutation imparts a very strong drive towards PD, resulting in full penetrance and an early onset of the disease, hence its impact is expected to diminish the effect of genetic background³.”

“Genetic background likely plays a greater role in PD caused by mutations with lower penetrance or in idiopathic cases. Therefore, in the future we will explore the potential overlap between the network identified in this study and the pathways altered by idiopathic disease, as well as the effect of genetic background, by investigating isogenic controls together with cell lines carrying PD mutations.”

2) Another important control lacking is the use of iPSC lines from idiopathic cases, this would be important to integrate into the analysis to confirm or refute some of the common PD mechanisms proposed. With such comparison one may infer a core network of PD proposed.

Answer: Dear reviewer, thank you for bringing up this important point. As a very first step towards identifying a core PD network, we wanted to focus on identifying a skeletal core network. Idiopathic cases can be caused by the interactions between genes of small effect and the environment, or environmental factors alone, broadening the spectrum of pathways involved in the development of PD in these cases. Many of these pathways are unlikely to show strong alterations, which may not be much greater than that of natural variation, hence difficult to distinguish from background noise generated by natural variation in unrelated pathways. Our approach is to first understand the effect of PD-associated mutations of strong effect, in order to detect pathways distinctly altered by such mutation, reducing the effect of background noise from natural genetic variation among individuals. Once this is established, the next important stage will be to include idiopathic cases and see whether they also map onto the same network.

The following passage was added to the discussion (lines 518-525).

“The challenge is that idiopathic cases can be caused by interactions between genes of small effect and the environment, or between environmental factors alone, which potentially broadens the spectrum of pathways involved in the development of PD in these cases^{4,5}. Many of these pathways are unlikely to be strongly altered by gene mutations and are likely difficult to distinguish from background noise generated by natural variation in PD-unrelated pathways. Therefore, we first focused to understand the effect of PD-associated mutations of strong effect, in order to detect a core network of pathways distinctly altered in PD.”

3) Data/photos showing the quality and the efficiency of the differentiation process into mDA neurons were lacking. Photos showing the differentiation / morphologies and efficiency of the mDA neuron differentiation and the quantitative analysis of the time course differentiation would be important to add.

Answer: As mentioned above as point 1 of the overall feedback, we extended the analysis of the differentiation process significantly and added new imaging, as well as transcription data to the manuscript in the new Fig. 3 and Fig. 4. To illustrate the time course and quality of the differentiations process, we used the mDA precursors and mDA neuron -specific gene expression described in a recent publication by Ásgrímsdóttir and Arenas (Figure 1 of their publication)¹. Using this information, we were able to document the appearance of mDA neuron precursors and of early mDA neurons at various timepoints of our differentiation protocol (Fig. 3d,e). The expression of mDA markers was confirmed by single cells analysis and by qPCR for TH, ALDH1A1 and LMX1A (Fig. 3b,c). Timing and trajectory of the differentiation process was further illustrated by staining for MAP, TH, LMX1A, PITX3 and DAT at day 25 and day 35 of the differentiation process (Fig. 3a). The expression of mDA markers by TH positive neurons was confirmed by co-staining of TH with LMX1A, PITX3 or DAT at day 35, in both control and PINK1 cell lines (Figure 4a). Additional ICC images were included in the Supplement Fig. 13).

4) Adding in vitro functional studies demonstrating the function of critical gene(s) in the iPSC model reported will further strengthen the conclusions.

Answer: To address the presence of possible functional deficits, we performed a proteomics analysis at D25 and D40 of the differentiation protocol and a new figure is dedicated to these results (Fig. 7). Our analysis showed altered levels of TH and DDC protein levels in cells carrying the PINK1 mutation, at both timepoints (D25 and D40), pointing to a dysregulation of dopaminergic pathways. Subsequent network analysis confirmed the involvement of PD-associated pathways and vesicular transport, replicated at both D25 and D40, as well as involvement of genes identified by SC-RNAseq (Fig. 7). Hence, the proteomics analysis strongly supports our previous transcriptional analyses. This validation approach is now added in the result and methods sections. Analyzing a broader spectrum of timepoints will be part of our future objectives.

The following passage was added to the Abstract (lines 13-14):

“Proteomics analysis showed a consistent alteration in proteins of dopamine metabolism, indicating a defect of dopaminergic metabolism in PINK1 neurons.”

The following passage was added to the Introduction section (lines 98-100):

“Subsequent proteomics analysis of differentiated cells confirmed the manifestation of the transcriptional modifications at the protein level.”

The following passage was added to the Results section (lines 335-349):

“Proteomics analysis confirms impaired neuronal phenotype in PINK1 mutant line

To investigate how the identified transcriptional modifications manifest in the neuronal phenotypes, we performed proteomics analysis at an early (day 25) and later maturation stage (day 40). The analysis identified 39 differentially abundant proteins in PINK1 cells as compared to controls, based on biological duplicates with a log₂ fold change larger than 1 (Fig. 7a). Of these, four differ at both timepoints (D25 and D40). Overall, 31 proteins were differentially abundant at D25, including CSRP2 and VWASA, which were also identified by sc-RNAseq as differentially expressed at the mRNA level at D6, D15 and D21 (Figure 7b, Supplement Table 13). At D40, 12 proteins were found to be differentially abundant,

including 4 also identified at D25, namely TH, DDC, NES and VIM. We performed a network analysis based on the differentially abundant proteins (Fig. 7b). The resulting network again connects PD-related nodes and exhibits a good overlap with the transcriptional-derived network. This consistent result indicates that the observed transcriptional modification led to an impaired neuronal phenotype, despite the subtle differences in expression, and further highlights the significance of the proposed PD Core network.“

The following passage was added to the Discussion section (lines 492-511):

“To investigate whether the observed transcriptional modifications lead to functional deficits that would further support the relevance of this model, we performed a proteomics analysis at D25 and D40 of the protocol. D21 represents early postmitotic neurons, while D25 represents early mature neurons and D40 mature neurons. Our first analysis showed dysregulation of dopaminergic metabolism at D25 and D40 of differentiation (Fig. 7b, Supplement Table 13). The list of differentially abundant proteins identified by proteomics analysis exhibits an overlap with the DEGs identified in the sc-RNAseq analysis (Fig. 7b, Supplement Table 13) and many of these proteins are already known to be involved in PD⁶. Importantly, two proteins that were differentially expressed at both D25 and D40, DDC and TH, are key enzymes closely associated with PD and are involved in dopamine metabolism^{7,8}. Altogether, four proteins were differentially expressed at both timepoints, in two independent biological replicates per timepoint (Fig. 7b, Supplement Table 13). The other two proteins are the cytoskeletal proteins VIM (Vimentin)⁹ and NES (Nestin), the latter is co-expressed with the PD-associated gene DJ-1 (PARK7)¹⁰. These were found to be abnormal also by other studies, and are involved in cytoskeletal transport, which represents a key aspect of PD pathology⁶. Performing a network analysis based on the proteome phenotype revealed a proteomics network related to the transcriptional network (Fig. 7b). In all, these data show a consistent abnormality in the levels of enzymes needed for DA metabolism, which indicates that cells carrying the PINK1 mutation have a functional deficit of the DA metabolic pathway that eventually can lead to neuronal loss of mDA.”

The description of techniques used was added to the Methods section (lines 776-807).

Reviewer #2.

Major

1) Page 6 – Did you gain control from ref28 and run the experiment with these controls or did use the control results from ref28? If it's the later, I will have further comments about the statistical analysis. From my reading of ref28 am I correct in thinking that you used the control cell lines in your experiment and performed the iPSC experiment in conjunction with the PIK1 mutants?

Answer: We thank the reviewer for pointing out the unclear description. We obtained the control as an iPSC cell line, the reference only provides information about the origin of the control iPSC cell line. Both control and PINK1 cell lines were differentiated at the same time, so that they would be subject to the same conditions. In order to minimize batch effects of the single cell analysis, samples from both cell lines were collected at the same time, processed, and sequenced using the same chip. We added the exact timing schedule as a supplementary

Table 2, since Figure 1 is quite general, and further clarified this point in the following section:

The following passage was added to the Introduction section (lines 87-90)

“We generated four pairs of samples, each consisting of a PINK1 and a control cell line differentiated in parallel. The pairs were differentiated in succession, so that they would reach collection day at the same time, yet at a different stage of differentiation (Fig. 1).”

The following passage was added to the Results section (lines 105-108, 202-210):

“We performed a systematic differential expression analysis at single-cell resolution between an iPSC line carrying the PD-associated ILE368ASN mutation in the PINK1 gene and an age- and sex-matched control cell line (control 1-2 in ref¹¹) **during their parallel differentiation** into mDA neurons (Fig. 1, Supplement Tables 1 and 2).”

“To investigate the effect of the *PINK1* mutation on mDA development, we differentiated the control and the PINK1 cell lines in **parallel** and focused on the early differentiation period, to increase our chances of finding the direct effects of PINK1 (Fig. 4). Co-staining of TH positive neurons with the midbrain dopaminergic markers PITX3, LMX1A and DAT in both the control and PINK1 cell lines identified neurons at day D21 as early postmitotic mDA neurons¹² with a clearly neuronal morphology and no major differences between the cell lines (Fig. 4a). To investigate potential underlying mechanisms of the PINK1 mutation, differential expression between the two, **in parallel differentiated**, cell lines at each time point was determined and genes that were identified as differentially expressed at all four timepoints were identified.”

2) Figures – Text in all figures should be legible, this includes all figures. It is difficult to determine the results when I am unable to even read the text on the figure. For example, figure 2b, between the resolution of the figure and the small text, none of this data can be interpreted.

Answer: We apologize for the inconvenience. We made significant revisions to the figures to ensure that the text is legible. We will also submit full resolution figures to avoid future issues.

3) Statistical tests and numbers are not clearly defined in the results or the figure legends. This makes interpretation of the data difficult as trying to determine if gene sets are previously known in the literature or derived from the analysis.

Answer: Dear reviewer, thank you for noticing this omission. The results of statistical tests were added. All data was generated, no results were drawn from literature. Most of the information was provided in the Method section and we now referenced this in the Result section and added the information to the figure legends where appropriate. The presented gene sets and networks all result from our data and analyses and are subsequently discussed in the context of literature findings.

4) Figure 5c D21 volcano plot does not look correct. This may be because of the maximum adjusted p-value statistic, over traditional methods, or this may be due to the gene directional change constraint that you imposed on the analysis, which is not appropriate for iPSC stage

analysis as gene expression fluctuate over the course of differentiation and so might the gene involved in PINK1.

Answer: We agree that the volcano plot in Figure 5c D21 had a strange form and we had tested different methods and revisited the analysis again based in the reviewer's comment in detail (see what is following). It is worth mentioning that here we only compared expression profiles between the cell lines at one time point and not between time points. Furthermore, we compared the profiles at stages which are expected to represent distinct differentiation stages of lineage commitment and therefore do not expect to observe significant effects of gene fluctuations, but more general changes corresponding to cell fate commitment. Based on this time point-wise comparison, we then subsequently investigated genes that were consistently differentially expressed at different time points as candidates for the PD network.

When revisiting the DEG analysis, we realized that the shown fold changes were the natural fold change retrieved from Seurat 3.1 and not the log₂ fold changes. This is corrected in the updated versions shown below but due to the additional analysis performed we decided to not present them in the manuscript any more for clarity reasons and to align with the typical representation of single cell RNA sequencing analysis.

For the volcano analysis, we tried different methods for differential expression analysis that led consistently to a similar results for the volcano, so the results seem to be consistent. For this analysis the adjusted-p-values of the time-point wise comparison between the samples at one time point and not across time points was used (which also holds true for the particular comparison between the Control and PINK1 samples as day d21). Therefore, the volcano plot is not based on the maximum p-value mentioned later in the manuscript. Hence, the unconventional shape of the updated volcano plot for D25 seems not to be based on an analysis artifact of DEG determination but to respect the properties of the single cell data where, specifically for low expression levels, the high number of replicates can lead to significant p-values. For space and clarity reasons these plots are not shown in the revised version of the manuscript.

We also followed the reviewer's comment and applied linear mixed effect models using GLMM to account for the time component and to compare the outcome with our results.

From the GLMM approach

```
formula <- paste("Treatment ~ ", feature, " + (1 | Timepoints)")
glmm <- lme4::glmer(formula, data = glmm_mat_f, family = binomial)
```

we selected the maximum fold changes of the different time points and the adjusted p-value (Bonferroni). For comparison with our results, we selected genes with $p_{adj} < 0.05$ and $\log_2FC > 1$ leading to the following volcano plot

and compared the DEGs identified by our approach with the GLMM approach:

The difference between the two methods was only 12 genes (<5%) where our approach identified 10 extra genes and missed two DEGs identified the GLMM method. Based on all this consistency checks, we decided to only show the overarching picture and, as suggested by the reviewer, to focus on the pathway analysis linked to the DEGs. The corresponding Figure (previously 5 now 6) has been accordingly modified.

5) Network analysis – Correlation of gene expression changes and protein level changes is low, which is further exacerbated by the low fold change cut off > 0.1. This is further emphasized by the statistical model gaining the list of differentially expressed genes. This means interpreting the network analysis is difficult. Rather than protein-protein interaction networks what did the traditional pathway network analysis show? You state an interaction with known PD genes, I am curious as to if you performed a KEGG analysis, is PD pathway the most highly associated pathway?

Answer: Dear reviewer, thank you for the suggestion to substantiate our network results by pathway analyses. We performed the KEGG pathway enrichment analysis using cluster Profiler. As input, we used all genes with Log2 Fold change > 0.5 and p_val_adj < 0.05 that were identified by the pairwise (timepoint-wise) differential expression analysis. The KEGG Parkinson's disease pathway was the top KEGG pathway associated with the differentially expressed genes forming the network (Fig. 5c) and it is highlighted in Supplement Figure 9a. CYCS, one of the genes of the KEGG Parkinson's pathway is a significant node of the network. Besides Parkinson's disease as the top association, three additional KEGG pathways identified were Spliceosome, Huntington's disease and Thermogenesis, in order of decreasing significance. We added KEGG pathway enrichment analysis to Figure 5 and details are listed in Supplement Table 12.

In respect to the fold change cut-off, we would like to comment that high fold change cut-offs are typically applied to reduce the number of differentially expressed genes and to focus on largest differences which can often indicate significant modifications. On the other hand, a large fold change does not necessarily have to imply a large effect on the downstream processes. Under some circumstances small fold changes can have a much larger impact than a large change (as e.g. in phosphorylation <https://link.springer.com/article/10.1186/s12964-019-0326-6>). For our analysis here where we compared paralleled differentiation between the 2 cell lines at the same time points, we would not necessarily expect large differences but more subtle changes because (i) cells follow the same differentiation protocol and should therefore in very comparable states, and (ii) the PINK1 cell line exhibits a loss of function mutation and not a knock-out mutation (which would be lethal) and therefore expression might not be impacted drastically (specifically in early stages). Hence, we used a less strict (FC based) strategy for detecting the DEGs, but subsequently identified and validated important and coherent biological signals by PPI networks and enrichment analysis.

Minor Introduction

6) Page 4 – “clearly plays a role in mitophagy”, “clearly” is not the correct language here, please state that multiple publications have shown PINK1s’ involvement in mitophagy. This is important as your next sentence states it has not been fully elucidated.

Answer: This has been corrected. The link to mitophagy was clarified and references 11-13 were added (lines 34-49).

7) Page 4 – “impossible to study them”, this is a highly controversial statement, as publications using microarray and even laser dissected RNAseq has been able and published to show you can study these. Please change the tone of this sentence to state that it has been difficult to study these neurons.

Answer: This has been corrected. We changed the sentence to “One of the main obstacles to the study of Parkinson’s disease is the death of mDA neurons and resulting shortage of available post-mortem samples.” (lines 51 – 52).

8) Page 4 – This comment continues as research was not only limited to animal model, please change.

Answer: This has been corrected. (lines 51 – 62)

9) Page 5 – Rather than stating “hence their study would not be as pertinent”, please state that using your model of mDA allows for the elucidation of mechanisms causing PD-induced cell death.

Answer: This has been corrected. (lines 65 – 67)

We changed the sentence to “We used an optimized differentiation protocol to specifically generate mDA neurons, as this cell type displays a unique susceptibility to cell death in PD¹²⁻¹⁴; the effect of PD on other types of DA neurons is variable^{15,16}.”

10) Page 5 – “which in turn leads to the classic movement symptoms”, which it turn has been directly associated to the classic movement symptoms.

Answer: Dear reviewer, thank you for pointing this out. This has now been corrected (lines 79 – 81).

“Their distinct identity is reflected in their function and current research indicates that this leads to their unique susceptibility to death in PD, which it turn has been directly associated to the classic movement symptoms of the disease¹⁴⁻¹⁸.”

11) Introduction – Could you please write one or two sentences about the mutation ILE368ASN, and any previous knowledge we know about this specific mutation. A short summary of the literature to better understand why you chose this mutation. Readers need to understand the logic of your decision to study this particular mutation. Is PINK1 even expressed in mDA, because you should mention this.

Answer: Dear reviewer, thank you for suggesting this. The following passage was added to the Introduction section (lines 37-44)

“The PINK1 protein is expressed ubiquitously throughout the brain, in all cell types, where it localizes to the mitochondrial membrane¹⁹. PINK1 is a mitochondrial ubiquitin kinase and, together with the cytosolic ubiquitin ligase PARKIN, it targets damaged mitochondria for degradation via mitophagy, performing a mitochondrial quality control function needed to prevent accumulation of damaged mitochondria, which otherwise results in neuronal cell death^{3,20,21}. The ILE368ASN mutation interferes with this process by abolishing PARKIN’s ubiquitin kinase activity through deformation of a binding pocket and substrate misalignment²⁰.”

Minor Results

12). Results – Please provide a sample identification supplementary table with sample information and timepoints used in the experiment. I may have missed it, but it would make reading this publication a lot easier. This would also help with which covariates you used in your analysis.

Answer: In addition to Figure 1, we added Supplementary Table 2 to provide this information. Additionally, the experimental design was clarified in the Results section (lines 105-111) by adding the following passage:

“We performed a systematic differential expression analysis at single-cell resolution between an iPSC line carrying the PD-associated ILE368ASN mutation in the PINK1 gene and an age- and sex-matched control cell line (control 1-2 in ref¹¹) during their parallel differentiation into mDA neurons (Fig. 1, Supplement Tables 1 and 2). After preprocessing and quality filtering, we used 4495 cells and 18,097 genes in our downstream analysis (Methods). For data integration, we performed a network analysis to identify the underlying key mechanisms of PD progression.”

13). Page 7 – “iPSC status was ascertained by standard methods”, please provide reference for this statement. “TaqMan iPSC Scorecard Assay”, again provide reference. None of these clearly described in the methods. Please update methods section.

A reference for Scorecard analysis was added and a references for Oct 3/4 and Tra-1-60 staining for pluripotency determination were also added on lines 117-120:

“The normal karyotype of iPSCs was confirmed (Supplement Fig. 2) and their iPSC status was ascertained by staining for the POU5F1 (also known as Oct4)²²⁻²⁵ and the TRA-1-80^{25,26} iPSC markers (Fig. 2a), as well as by a TaqMan iPSC Scorecard Assay^{27,28} which also confirmed the trilineage potential of the cell line²⁷ (Fig. 2c).”

A reference to a review which describes the procedures used for pluripotency determination was added on lines 124-125:

“..... approaches for determining iPSC status (reviewed by Smith and Stein)²⁵.”

We added references to the use of MYC and POU5F1 as pluripotency indicators on lines 130-132:

“In our dataset we quantified the expression of genes commonly used to ascertain iPSC status (MYC46 and POU5F139–42) and showed that these can be readily detected by sc-RNAseq analysis (Fig. 2b).

More detail was added to Methods section: lines 597-614

14). Page 7 – Figure 3a Where did the gene to confirm characterization of iPSCs expressed genes come from? Do you have a reference? Or are these differentially expressed genes between the timepoints of the controls? If they are why do you not state, the model you used? This is also not stated in the figure legend.

Answer: The relevance to stemness of genes selectively expressed in iPSCs and detected by differential expression analysis between timepoints of our single cell RNA seq analysis data (Supplement Fig. 12), is now supported by references. In short, we looked at the genes specifically expressed only in the iPSC group, as compared to differentiating cells, and showed that, based on published research, these genes are indicators of stemness.

The following correction and addition of references was performed (lines 140-143):

“For instance, stem cells with the highest expression of stemness markers²⁴ express TDGF-1. Additional genes expressed by the iPSCs were L1TD1, USP44, POLR3G, and TERF1 (essential for the maintenance of pluripotency in human stem cells^{29–32}), as well as IFITM1, DPPA4, and PRDX1 (associated with stemness^{33–35}).

It is useful to create a list of genes associated with various states, such as mDA status or high stemness, which are readily detectable in SC-RNAseq data, creating a translation between iPSC state characterized by ICC, Scorecard or qPCR and an equivalent signature visible in SC-RNAseq data. Such list then can be used by others to check for stemness in the SC-RNAseq data, where additional experiments are not possible, such as when using sc-RNAseq data from a databank.

15). Page 8 – This part was well written and concise. The only change is to clearly state the model, are you comparing only controls across differentiation states? Are you comparing all samples and regressing out the PINK1 effects in the model? Or more likely is this just average expression across all the controls?

Answer: In the newly added Figure 3 we now present data from the differentiation of the control cell line over the full differentiation process extended to D35 and D50, but not for the PINK1 cell line, since we first used only the control cell line to explore the differentiation process in more detail before investigating the modifications triggered by the PINK1 mutation. Hence, the description is only based on the control cell line and no regression of PINK1 effects was performed. The comparison between the cell lines is now shown in the extended Fig. 4, which now includes staining images of both the control and PINK1 cell lines in parallel, to illustrate that both cell lines differentiated with the same efficiency and expressed the same mDA markers.

16). Page 8 – It is surprising that you only identified so few differentially expressed genes. Was there unwanted variation that could account for this? Did you try any unwanted variation removal? Were there any multicollinearity issues with the data? With such a complicated question, did you consider linear mixed effect models to account for the time component?

Answer: As mentioned above (response to Major point 4), we are comparing expression between the cell lines always at the same time point and not across the different time points. Therefore, we do not expect super drastic changes since the overall developmental dynamics is rather similar and the mutation has more subtle effects. This might be also related to the compensatory effect in patients which often only exhibit symptoms above the age of 50.

To further validate our results, we followed your suggestion and used the MAST method and the total number of transcripts as a covariate to remove any unwanted variation originating for technical reasons. The applied linear mixed models are explained above in relation to Major Point 4.

```
pbmc.markers <- FindMarkers(object = object,  
                             ident.1 = ident.1,  
                             ident.2 = ident.2,  
                             assay = "RNA", min.pct = 0.1,  
                             logfc.threshold = 0.0,  
                             only.pos = FALSE,  
                             test.use = "MAST", latent.vars = c("nCount_RNA"))
```

This analysis resulted in very similar gene set as the one obtained by our approach which indicates the robustness of the findings.

17). Page 9:10 – I need literature justification of a foldchange of > 0.1 being biological important. With the low number of differentially expressed genes, using a maximum adjusted p-value may not be appropriate as genes are going to change between iPSC stages. Limiting the analysis to genes that have to change in the same direction is not biologically sound as iPSC conversion is not a linear gene expression change. Genes fluctuate across stages and so would disease genes. This also may explain the strange volcano plot of D21 in Figure 5c. Data integration reveals a common PD network – With the questions about the statistical tests, maximum adjusted p-value and the gene direction constraints I will not go into too much detail about the PPI. I believe the data and the overall PPI results are correct, and this part is well written. The constraints to gene direction and statistics should be reassessed. I think doing a simple traditional pathway analysis, using ranked based gene expressed (GSEA), would alleviate a lot of the questions I have about how you came to the list of differentially expressed genes.

Answer: As explained in much more detail above (in relation to Major Point 4 and the volcano plot), we compare expression profiles at the same time point and expect also more subtle modifications due to the mutations. Furthermore, we investigate expression profiles at major developmental stages of cell lineage commitment where we do not expect fluctuating expression. Hence, a low cut-off for the fold changes represents a more sensitive choice to identify the potential subtle differences between the cell lines. The above shown complementary analysis with mixed linear models and the now included pathway analysis are additional indication for the robustness of our results and the relevance of low fold changes. Furthermore, the added proteomics analysis to validate the phenotypic consequences (shown in the new Fig. 7) are also demonstrating that the expression-based network affects the phenotype. Based on these complementary results, we believe that our strategy is sufficient to identify underlying mechanisms even with subtle changes in expression.

Minor Methods

18). Statistics – Title should be qPCR statistics, you should also mention the actual statistics used to determine significance and you should mention which method you used (delta delta ct) and you need to mention how you normalized to housekeeping genes (name them) and how many housekeeping genes you used.

Answer: Thank you for the comment, this has now been done. The passage (lines 654-670) was changed to the following:

“qPCR statistics

In real-time qPCR graphs, each timepoint consists of three independently differentiated samples, seeded at the same time, hence representing biological replicates. The sample concentration was determined by absolute quantitation, comparing the sample concentration to a known concentration of a standard template identical to the one being amplified. The value was standardized to total RNA, as cDNA was synthesized from a standard amount of total RNA for each sample. This value was then divided by the concentration of GAPDH obtained for that particular sample, thus standardizing to GAPDH levels and generating a unitless number denoting expression relative to the expression of the housekeeping gene GAPDH. GAPDH was selected from among a number of possible housekeeping genes, as it showed the best ability to normalize gene expression in a population of untreated samples (this approach was previously described for qPCR experiments³⁶). Each of the samples was amplified in duplicate. Each sample value was an average of the experimental duplicate. Standard error was calculated as standard deviation of the three biological replicates (each a result of experimental duplicate), divided by the square root of the number of samples. This method is described in more detailed in Novak et al.³⁷”

19). Single-cell RNA-seq data analysis – Please provide a reference to the quality filtering parameters. These should be the same as the standard for Dropseq and please provide reasoning for any that are not standard in the literature.

Answer: For the analysis, a subset of cells with the highest cumulative expression was considered for the analysis (see e.g. step-by-step workflow for low-level analysis of single-cell RNA-seq data with Bioconductor and <https://pubmed.ncbi.nlm.nih.gov/27909575/>) in agreement with the DropSeq standard. CellRanger also follows this strategy to prefilter the 10x Data. They order the cells based on the UMI counts and then try to find the transition to pre-filter low quality cells (see e.g. <https://support.10xgenomics.com/single-cell-gene-expression/software/pipelines/latest/algorithms/overview>)

In addition to this filtering, we defined cells as low-quality based on three criteria for each cell in accordance with the standard DropSeq procedure. The number of expressed genes must be more than 200 and 2 median-absolute- deviations (MADs) above the median; the total number of counts has to be 2 MADs above or below the median, and the percentage of counts to mitochondrial genes has to be 1.5 median-absolute- deviations (MADs) above the median.

20). Single-cell RNA-seq data analysis – For the MAST analysis did you use a contrast matrix approach to weight the variability of the entire experiment, if not why? Also, which covariates did you adjust for in your model?

Answer: As explained above (Minor Comment for Page 8), we used the MAST method and the total number of transcripts as a covariate to remove any unwanted variation coming from technical reasons.

21). Single-cell RNA-seq data analysis – Fold changes > 0.1 is not a standard in the literature, can you please provide a reference where this has been implemented.

Answer: Please see our responses above (in particular the the Major Point 4).

22). Data availability – For constancy in bioinformatics approaches, many high impact journals now require github repositories for all code used the analysis. This is important to both clarity in analysis and reproducibility, furthermore I would not need to ask basic statistical questions in my review process such as contrast matrices. Are you going to make your code available and if not justify your reason?

Answer: Yes, we will make the code as well as the data available once the manuscript is accepted, as we typically do for our publications. For the code the corresponding gitlab is already setup https://git-r3lab.uni.lu/dimitrios.kyriakis/ipses_pink1/ and will become public upon acceptance and data will be uploaded to GEO (as done for our previous single cell data).

Minor Figures

23). Figure 1. – Place numbers of replicates in figure legend

Answer: This has now been done. The following sentence was added to Figure 1 legend: “The gene expression matrix consists of 4495 cells”

24). Figure 2. – Text larger, current figure text is illegible. What stage of differentiation done was the IHC? What was the sample size?

Answer: Dear reviewer, thank you for pointing out that this figure was not clear. We have clarified the caption for this figure by adding the following text: “Staining for the iPSC markers OCT3/4 and TRA-1-60 of iPSC colonies, prior to differentiation. DAPI was used to stain cell nuclei as a reference.”

Figure text was revised to correct legibility issues.

We also added the following clarification to the Methods section (lines 598-602):

A 24-well cell culture plate was seeded with iPSCs, one or two wells per cell line, and the iPSCs were then allowed to form colonies. At least a dozen colonies were present in each well and images were taken of several representative stained colonies. This was performed prior to any major experiment, to confirm the status of the cell line. Any evidence of differentiation identified by a loss of iPSC marker expression would be documented.

25). Figure 3. – Text larger, current figure text is illegible. Again, provide n number for experiments and the statistical tests. Is heatmap a unsupervised clustered? Is this differentially expressed genes or a predetermined list from a publication in 3a? You also need to put models in either the text or the figure legend to make this clear. Also, for the qPCR, what housekeeping genes did you use, what statistical test. All these things should be clearly stated.

Answer:

Dear reviewer, this figure was completely replaced. We now show expression of only select genes, which allowed for larger labels (Figures 1, 2 and 3).

The following statement was added to the figure legend: “The gene expression matrix consists of 4495 cells”

26). Figure 4. – Text larger, however this figure is just legible when zoomed in. I would also like to know if you only used controls for this analysis.

Answer: Dear reviewer, this figure was almost completely replaced by a new set of figures, Figure 3 and Figure 4. In Figure 3 we used the control cell line to illustrate the differentiation process and this is now described in detail in the text:

“The expression was confirmed using our single-cell RNAseq data for the control cell line (Fig. 3b), since this dataset contained additional datapoints (D10, D26, D35 and D50).”

In Figure 4 we illustrate the onset of expression of mDA markers in both cell lines (control and PINK1), first at the protein level, by immunocytochemistry (Fig. a), and second, by gene expression using our SC-RNAseq data (Fig. 4b).

27). Figure 5. – Text larger, statistical models and tests need to be supplied in the figure legend. Also the sample numbers need to be supplied. Figure 5c volcano plot need to be reassessed.

Answer: Dear reviewer, we increased the text as much as possible and we will upload a high resolution figure. This figure was completely revised.

28). Figure 6. – Completely illegible.

Answer: Dear reviewer, this figure has also been completely revised and new graphs were generated, making sure that the labels are as legible as possible.

Reviewer #3 (Remarks to the Author):

This manuscript combines single cell transcriptomic approaches with the generation of midbrain dopaminergic neurons obtained from human induced pluripotent stem (iPSc) from patients with Parkinson’s disease (PD), to report unified networks underlying sporadic and mutations driven pathology. The manuscript is organized and well written. This research will provide a great resource for other groups exploring the next steps for treatment and characterization of PD. It was a pleasure to read this important manuscript.

At the same time, my only major comment is that the data presented in this manuscript are largely descriptive. Although substantial computational predictions were provided, direct impact of DEGs on cellular state and neuronal properties was not further tested. Causality of the interactions needs to be demonstrated. Moreover, whether abnormal gene expression impairs key PD-related mechanisms or pathways, including mitochondrial, protein processing or RNA metabolism dysfunction, need to be assessed and demonstrate.

I offer below a list of suggestions and comments

1). What are the evidences (beside abnormal gene expression) that these iPSCs derived mDA neurons are **indeed recapitulating some of the PD pathophenotype**? The authors could utilize their model and demonstrate by IHC or RNA in situ hybridization, temporal and spatial expression of these genes. More importantly, it will be interesting to reveal new abnormal neuronal properties (early onset) that arise prior to neuronal death (i.e. oxidative stress, DNA breaks, mitochondrial dysfunction, mislocalization of protein)

2). In the discussion, the authors offer a list of genes (such as LGI1, CHCHD2) which “had no known connection to molecular mechanisms underlying PD pathology, though some of them are known to be involved in sporadic PD”. Since I think this is one of their major findings, the authors should pick one or two of these pathways and further demonstrate the causality of interactions with PD-related mechanisms. For example, LGI1-CNTNAP2 interaction could be further validated by examination of axon organization.

Minor comments:

3). Network figures (Fig 6. and Supplementary Figure 11.) are impossible to see, please revise - Pg. 17 Line 365, “though some of them are known”

Answer: Dear reviewer, thank you for your very valuable and insightful feedback.

1). Recapitulation of some aspects of a PD pathophenotype.

In order to explore the possibility of a manifestation of an abnormal neuronal phenotype, we initiated a collaboration with a group performing proteomics and together we were able to show that *in-vitro* generated neurons carrying the PINK1 mutation show dopaminergic metabolism deficits. Our experiment showed that the DDC and TH are abnormally expressed at the protein level, both at day 25 and day 40 of the differentiation protocol. In total, four proteins were abnormally expressed in the PINK1 neurons, at both day 25 and day 40, indicating a dysfunction in dopamine metabolism (Fig. 7). Hence, the proteomics analysis indicates that the identified differentially expressed genes do actually manifest in impaired protein phenotypes with an overlap to the sc-RNAseq data. Also, the corresponding protein-based pathway and network analyses exhibit phenotypic traits related to PD despite the early time points.

We performed additional comprehensive imaging to characterize differentiation dynamics and resulting neuronal phenotypes. This analysis further demonstrates the mDA specificity of our differentiation protocol (newly added Fig. 3). The comparison between CTRL and PINK1-mutation cell lines (Fig. 3 and Fig.4) did not exhibit a significant difference which is probably due to the early time point at which modifications may not have manifested morphologically. Together with the identified molecular impairment by sc-RNAseq and proteomics this may further indicate the developmental aspect of PD progression where early minor modifications will establish only later a severe phenotype.

2). We agree with the reviewer that such an investigation would be very interesting. While we do observe an effect on the protein level, our imaging results did not exhibit a clear phenotypic difference probably due to the early time points we studied here (as described above). Hence, we would expect that the molecular impairment will only later establish a morphological and more severe physiological phenotype. Since the focus of the current

manuscript is to investigate the early differentiation and the by the mutation triggered (subtle) modifications, we will follow up these hypotheses in future work.

3). These figures were entirely revised, new graphs were created using Gephi and care was taken to ensure that the labels are readable. The missing space on line 365 was added.

References mentioned in the text:

1. Ásgrímsdóttir, E. S. & Arenas, E. Midbrain Dopaminergic Neuron Development at the Single Cell Level: In vivo and in Stem Cells. *Frontiers in Cell and Developmental Biology* (2020) doi:10.3389/fcell.2020.00463.
2. Zanon, A., Pramstaller, P. P., Hicks, A. A. & Pichler, I. Environmental and genetic variables influencing mitochondrial health and Parkinson's disease penetrance. *Parkinson's Disease* (2018) doi:10.1155/2018/8684906.
3. Schneider, S. A. & Klein, C. *PINK1 Type of Young-Onset Parkinson Disease*. *GeneReviews*® (1993).
4. Chen, H. & Ritz, B. The search for environmental causes of Parkinson's disease: Moving forward. *Journal of Parkinson's Disease* (2018) doi:10.3233/JPD-181493.
5. Gwinn, K. *et al.* Parkinson's disease biomarkers: perspective from the NINDS Parkinson's Disease Biomarkers Program. *Biomark. Med.* **11**, 451–473 (2017).
6. Clark, E. H., Vázquez de la Torre, A., Hoshikawa, T. & Briston, T. Targeting mitophagy in Parkinson's disease. *J. Biol. Chem.* **296**, 100209 (2021).
7. Burkhard, P., Dominici, P., Borri-Voltattorni, C., Jansonius, J. N. & Malashkevich, V. N. Structural insight into Parkinson's disease treatment from drug-inhibited DOPA decarboxylase. *Nat. Struct. Biol.* **8**, 963–7 (2001).
8. Tabrez, S. *et al.* A synopsis on the role of tyrosine hydroxylase in Parkinson's disease. *CNS Neurol. Disord. Drug Targets* **11**, 395–409 (2012).
9. Siciliano, R. A. *et al.* Decreased amount of vimentin N-terminal truncated proteolytic products in parkin-mutant skin fibroblasts. *Biochem. Biophys. Res. Commun.* **521**, 693–698 (2020).
10. Yan, H. & Pu, X.-P. Expression of the Parkinson's Disease-Related Protein DJ-1 during Neural Stem Cell Proliferation. *Biol. Pharm. Bull.* **33**, 18–21 (2010).
11. Schöndorf, D. C. *et al.* iPSC-derived neurons from GBA1-associated Parkinson's disease patients show autophagic defects and impaired calcium homeostasis. *Nat. Commun.* (2014) doi:10.1038/ncomms5028.
12. Arenas, E., Denham, M. & Villaescusa, J. C. How to make a midbrain dopaminergic neuron. *Dev.* (2015) doi:10.1242/dev.097394.
13. Kriks, S. *et al.* Floor plate-derived dopamine neurons from hESCs efficiently engraft in animal models of PD. *Nature* (2012) doi:10.1038/nature10648.Floor.
14. Hegarty, S. V., Sullivan, A. M. & O'Keefe, G. W. Midbrain dopaminergic neurons: A review of the molecular circuitry that regulates their development. *Developmental Biology* (2013) doi:10.1016/j.ydbio.2013.04.014.
15. Anderegg, A., Poulin, J. F. & Awatramani, R. Molecular heterogeneity of midbrain dopaminergic neurons - Moving toward single cell resolution. *FEBS Letters* (2015) doi:10.1016/j.febslet.2015.10.022.
16. Giguère, N., Nanni, S. B. & Trudeau, L. E. On cell loss and selective vulnerability of neuronal populations in Parkinson's disease. *Frontiers in Neurology* (2018) doi:10.3389/fneur.2018.00455.
17. Blaess, S. & Ang, S. L. Genetic control of midbrain dopaminergic neuron development. *Wiley Interdisciplinary Reviews: Developmental Biology* (2015) doi:10.1002/wdev.169.
18. Roeper, J. Dissecting the diversity of midbrain dopamine neurons. *Trends in Neurosciences* (2013) doi:10.1016/j.tins.2013.03.003.
19. Gandhi, S. PINK1 protein in normal human brain and Parkinson's disease. *Brain* **129**, 1720–1731 (2006).
20. Ando, M. *et al.* The PINK1 p.I368N mutation affects protein stability and ubiquitin

- kinase activity. *Mol. Neurodegener.* **12**, 32 (2017).
21. Rakovic, A. *et al.* PINK1-dependent mitophagy is driven by the UPS and can occur independently of LC3 conversion. *Cell Death Differ.* **26**, 1428–1441 (2019).
 22. Pesce, M. & Schöler, H. R. Oct-4 : Gatekeeper in the Beginnings of Mammalian Development. *Stem Cells* **19**, 271–278 (2001).
 23. Niwa, H., Miyazaki, J. & Smith, A. G. Quantitative expression of Oct-3/4 defines differentiation, dedifferentiation or self-renewal of ES cells. *Nat. Genet.* **24**, 372–376 (2000).
 24. Hough, S. R., Laslett, A. L., Grimmond, S. B., Kolle, G. & Pera, M. F. A continuum of cell states spans pluripotency and lineage commitment in human embryonic stem cells. *PLoS One* (2009) doi:10.1371/journal.pone.0007708.
 25. Smith, K. P., Luong, M. X. & Stein, G. S. Pluripotency: Toward a gold standard for human ES and iPS cells. *J. Cell. Physiol.* **220**, 21–29 (2009).
 26. Bhattacharya, B. *et al.* Gene expression in human embryonic stem cell lines: unique molecular signature. *Blood* **103**, 2956–2964 (2004).
 27. Tsankov, A. M. *et al.* A qPCR ScoreCard quantifies the differentiation potential of human pluripotent stem cells. *Nat. Biotechnol.* **33**, 1182–1192 (2015).
 28. Bock, C. *et al.* Reference Maps of Human ES and iPS Cell Variation Enable High-Throughput Characterization of Pluripotent Cell Lines. *Cell* **144**, 439–452 (2011).
 29. Emani, M. R. *et al.* The L1TD1 protein interactome reveals the importance of post-transcriptional regulation in human pluripotency. *Stem Cell Reports* (2015) doi:10.1016/j.stemcr.2015.01.014.
 30. Lund, R. J. *et al.* RNA Polymerase III Subunit POLR3G Regulates Specific Subsets of PolyA+ and SmallRNA Transcriptomes and Splicing in Human Pluripotent Stem Cells. *Stem Cell Reports* (2017) doi:10.1016/j.stemcr.2017.04.016.
 31. Liu, Q. *et al.* The miR-590/Acvr2a/Terf1 Axis Regulates Telomere Elongation and Pluripotency of Mouse iPSCs. *Stem Cell Reports* (2018) doi:10.1016/j.stemcr.2018.05.008.
 32. Suresh, B., Lee, J., Kim, H. & Ramakrishna, S. Regulation of pluripotency and differentiation by deubiquitinating enzymes. *Cell Death and Differentiation* (2016) doi:10.1038/cdd.2016.53.
 33. Fu, Y. *et al.* IFITM1 suppresses expression of human endogenous retroviruses in human embryonic stem cells. *FEBS Open Bio* (2017) doi:10.1002/2211-5463.12246.
 34. Madan, B. *et al.* The Pluripotency-Associated Gene Dppa4 Is Dispensable for Embryonic Stem Cell Identity and Germ Cell Development but Essential for Embryogenesis. *Mol. Cell. Biol.* (2009) doi:10.1128/mcb.01970-08.
 35. Kwon, S. C. *et al.* The RNA-binding protein repertoire of embryonic stem cells. *Nat. Struct. Mol. Biol.* (2013) doi:10.1038/nsmb.2638.
 36. Novak, G. & Talerico, T. Nogo A, B and C expression in schizophrenia, depression and bipolar frontal cortex, and correlation of Nogo expression with CAA/TATC polymorphism in 3'-UTR. *Brain Res* **1120**, 161–171 (2006).
 37. Novak, G., Fan, T., O'Dowd, B. F. & George, S. R. Striatal development involves a switch in gene expression networks, followed by a myelination event: implications for neuropsychiatric disease. *Synapse* **67**, 179–88 (2013).

Reviewers' comments:

Reviewer #1 (Remarks to the Author):

The revised version of the manuscript has been extended with the addition of proteomic analysis for DA differentiation days 25 and 40. However some of the crucial controls requested in the first reviews are still lacking. The manuscript remains purely descriptive of the transcriptional and partial protein expression profiles, which are quite routinely reported now in many systems as part of cell characterization. This study needs functional validation (such as CRISPR-mediated for example) of the potential involvement of the candidate pathways/genes in the PINK1-mediated PD pathologies. The quality of the immunostaining in figures 3 and 4 still quite low, with high background / unspecific staining – The author may need to pay careful attention to the TH+ versus MAP2+: In figure 3a (MAP2/TH), TH+ cells show small neurite outgrowth – while MAP2 always present extended neurite outgrowth/branching (red). However, in Figure 4, expect for panel (a) TH/LMX1a, the TH+ look lot like MAP2 red+ with extended neurites. Would be helpful for reviews to label the figures with their respective numbers.

Reviewer #2 (Remarks to the Author):

The publication "Single-cell transcriptomics of human iPSC differentiation dynamics reveal a core network of Parkinson's disease", is a very well-written publication. I would like to thank the reviewer for thoroughly going over both my major and minor edits. These authors did a good job answering most of my comments and particularly adding further clarification to not only my questions but extending this in the publication. I thoroughly critiqued this publication and am very satisfied with the changes added to the publication. I have no further comments and am excited to see this being published, as this will be a great resource for both Parkinson's Disease and neurodegeneration. I look forward to their future work in Idiopathic IPSCs.

Reviewer #3 (Remarks to the Author):

The authors have done an excellent job revising the manuscript and addressing all reviewer comments. I believe the paper is acceptable for publication in Communications Biology.

Specifically, I think the proteomics data and the correlation with the scRNA-seq data, significantly increases the importance and the relevance of this iPSC model, for PD research. Identification of early deficits in proteins, have the potential to become biomarker for the onset of the disease, thus are critically important for this research.

With that, I agree that "time" is a significant and important factor in studying neurodegenerative diseases, and poses a challenge when with iPSC models. Indeed, it seems possible that the authors selected an early time point, when significant physiological aberration and still not present. Hence, it will be nice to mentioned those limitations in the discussion part. Also, it will be very interesting in future research to follow these cell lines for longer time periods, to introduce the "aging" factor as well.

I also liked the authors, side by side analysis, of DEGs identified with 2 different approaches (with or without the GLMM). I think the relatively large overlap between the datasets, strengthen their results, and thus could be also introduce in the text/supplements.

Dear Reviewers,

We would like to thank you for your valuable feedback.

To address the remaining comments, please see below our responses and the corresponding changes to the manuscript.

Reviewer #1.

1. Targeted analysis

We do not simply report here a minimal set of necessary cell characterisation experiments but rather chose to focus on single cell analysis of the iPSC cell line as a baseline for analyzing changes associated with its differentiation. This approach allowed us to validate and quantify the differentiation protocol and, hence, its relevance for the study that followed. Furthermore, the parallel sc-RNAseq time course data of the mutation and control cell line goes beyond a descriptive approach since it allows for sensitive identification of early transcriptome modifications which are typically not observed in purely descriptive approaches. Together with the proteomics data, this analysis enabled a cross validation of the network candidate based on independent biological replicates and methodologies. A full set of cell characterization experiments for this cell line is far broader than presented here and is included in a Lab resource manuscript submitted to the Stem Cell Research journal.

2. CRISPR approach

At this stage we did not include a CRISPR-corrected or a CRISPR-modified cell line. Instead, we used the experimental design to provide a filter aimed at the retention of pathways associated with the presence of the mutation to reveal early changes in expression and proteome. As can be noted from Figure 4b, there is a significant change in transcriptome between different differentiation stages, so identifying only pathways altered at all timepoints eliminated differentiation-associated pathways. On the other hand, both cell lines expressed a very homogenous transcriptome at each timepoint and this allowed us to eliminate common pathways for each timepoint. The underlying hypothesis is that pathways altered above this baseline will be associated with the PD mutation. The fact that nearly 70% of DEGs identified are already linked to PD and that a vast majority of DEGs interact to form a network supports the conclusion that a large proportion of the identified DEGs are in fact associated with the presence of this PD mutation. Together with the independent proteomics data (see reply above) and corresponding overlap, the analysis reveals an underlying network based on biological functionality which goes beyond a monogenetic perspective and can provide an important resource for the community to interpret results from diverse model systems.

We are currently working on a study which also includes CRISPR-corrected cell lines, but it should be kept in mind that the effect of a PD mutation, or almost any mutation in that matter, is often complex as further indicated by our here identified network. The inclusion or correction of a mutation does not automatically lead to the identification of pathways causal for the pathogenicity but may yield many pathways that are affected, but not necessarily linked to the disease in any way. Hence, it is important to use several pairs of cell lines and look at an overlap, where non-isogenic comparisons can reveal important pathways and isogenic comparison allow for characterization of the genetic background.

2. Imaging quality.

We reviewed the images and realized that they did not correctly represent the appearance of the originals, some of this effect may be caused due to dimming as a result of file type conversion. The tiff files generated from the originals, or new, clearer images, are now included in Figure 3 and 4. We hope that this does address the issue. We also added instrument and software details on lines 627-631, in the Methods section. All images are original, acquired using the same settings of the Zen software, only adjusting saturation. The tiff file and scale bar were then generated using Fiji.

The mentioned difference between Fig. 3 and Fig. 4 may be due to the fact that in Figure 3 the first row of images is of D25, early neurons, while in Figure 4 all the images are of D35 neurons, which are considerably more mature. We have gone back to the original files and exported them from the correct group of images, to make sure that no substitution has occurred. It is possible that some TH and MAP2 neurons do not appear identical, as it was noted by us and also by other groups¹, that not all TH+ neurons are MAP2+ and may represent a different subgroup. We did notice that TH neurons do not necessarily express MAP2, but almost all TH+ neurons in our experiment express mDA markers, while not all MAP2 neurons do.

Reviewer #3.

1. Limitations of using an early timepoint and future to follow these cell lines for longer time periods, to introduce the “aging” factor as well.

Dear reviewer, we now clarified this subject in the discussion (lines 360-364, 369-376) where we clarify that the use of an early timepoint enabled us to address two potential confounding factors. Firstly, we wanted to avoid detecting pathways altered due to stress, such as from mitochondrial death or metabolic stress due to poor mitochondrial function. By selecting an early timepoint, knowing that death occurred mainly in mature or aged neurons, we aimed to focus on alteration predating the response to overwhelming damage. Secondly, by focusing only on pathways altered at all timepoints, we aimed to eliminate pathways associated with the differentiation process, as each timepoint was from a very different stage and contained precursors which were quite dissimilar from each other, as shown in Figure 4b, but quite similar to the other cell line, also apparent in Figure 4b. The latter characteristic was aimed at detecting PINK1-mutation associated changes by pairwise comparison, now identifying genes not only expressed independently of cell state, but also expressed above this homogeneous background.

Based on our here presented data, we are now in the process of generating data to analyse changes associated with aging. For that purpose, we have identified genes, splice variant ratio or expression profiles of which are known to change with age in-vivo. We were able to show that this is recapitulated in-vitro after approximately D40 to D50 in culture. The preliminary data is also confirming the here identified network and we are currently investigating, using additional cell lines, how aging is emphasizing mitochondrial pathway pathology.

We, again, thank the reviewers for their very valuable feedback and hope that they agree with the modified version of the manuscript.

Reference:

1. Cossette, M., Lévesque, D. & Parent, A. Neurochemical characterization of dopaminergic neurons in human striatum. *Parkinsonism Relat. Disord.* **11**, 277–286 (2005).